



# Exploration of the atmospheric chemistry of nitrous acid in a coastal city of southeastern China: Results from measurements across four seasons

**Baoye Hu[1,2,3,4], Jun Duan[6], Youwei Hong[1,2], Lingling Xu[1,2], Mengren Li[1,2], Yahui Bian[1,2], Min Qin[6*], Wu Fang[6], Pinhua Xie[1,5,6,7], Jinsheng Chen[1,2*]**

[1]Center for Excellence in Regional Atmospheric Environment, Institute of Urban Environment, Chinese Academy of Sciences, Xiamen 361021, China

[2]Key Lab of Urban Environment and Health, Institute of Urban Environment, Chinese Academy of Sciences, Xiamen 361021, China

[3]Fujian Provincial Key Laboratory of Pollution Monitoring and Control, Minnan Normal University, Zhangzhou, 363000, China

[4]Fujian Provincial Key Laboratory of Modern Analytical Science and Separation Technology, Minnan Normal University, Zhangzhou, 363000, China

[5]University of Chinese Academy of Sciences, Beijing 100086, China

[6]Key Laboratory of Environment Optics and Technology, Anhui Institute of Optics and Fine Mechanics, Chinese Academy of Sciences, Hefei, 230031, China

[7]School of Environmental Science and Optoelectronic Technology, University of Science and Technology of China, Hefei, 230026, China

*Correspondence to*: Jinsheng Chen (jschen@iue.ac.cn) &Min Qin (mqin@aiofm.ac.cn)

**Abstract.** Because nitrous acid (HONO) photolysis is a key source of hydroxyl (OH) radicals, identifying the atmospheric sources of HONO is essential to enhance the understanding of atmospheric chemistry processes and improve the accuracy of simulation models. We performed seasonal field observations of HONO in a coastal city of southeastern China, along with measurements of trace gases, aerosol compositions, photolysis rate constants ($J$), and meteorological parameters. The results showed that the average observed concentration of HONO was $0.54 \pm 0.47$ ppb. Vehicle exhaust emissions contributed an average of 1.64 % to HONO, higher than the values found in most other studies, suggesting an influence from diesel vehicle emissions. The mean conversion frequency of $NO_2$ to HONO in the nighttime was the highest in summer due to water



droplets was evaporated under the condition of high temperatures. Based on a budget analysis, the rate of emission from
unknown sources ($R_{unknown}$) was highest around midday, with values of 4.35 ppb·h$^{-1}$ in summer, 3.53 ppb·h$^{-1}$ in spring,
3.13 ppb·h$^{-1}$ in autumn, and 2.05 in winter. Unknown sources made up the largest proportion of all sources in summer
(78.55 %), autumn (71.51 %), spring (69.67 %), and winter (55.63 %). The photolysis of particulate nitrate was probably a
source in spring and summer while the conversion from $NO_2$ to HONO on BC enhanced by light was perhaps a source in
autumn and winter. The variation of HONO at night can be exactly simulated based on the HONO/$NO_x$ ratio, while the
$J(NO_3^-\_R) \times pNO_3^-$ should be considered for daytime simulations in summer and autumn, or $1/4 \times (J(NO_3^-\_R) \times pNO_3^-)$ in
spring and winter. Compared with $O_3$ photolysis, HONO photolysis has long been an important source of OH except for
summer afternoon. Observation on HONO across four seasons with various auxiliary parameters improves the
comprehension of HONO chemistry in southeastern coastal China.

## 1 Introduction

Nitrous acid (HONO) photolysis produces hydroxyl  radical (OH), an important oxidant, in the troposphere (Zhou et al.,
2011). OH plays an important role in triggering the oxidation of volatile organic compounds and therefore determine the fate
of many anthropogenic atmospheric pollutants (Lei et al., 2018). Recent research results have shown that HONO production
is the cause of an increase in secondary pollutants (Li et al., 2010; Gil et al., 2019; Fu et al., 2019). Though extensive studies
have been conducted in the four decades since the first clear measurement of HONO (Perner and Platt, 1979), the HONO
formation mechanisms are still elusive, especially during the daytime, when there is a large difference between measured
concentrations and those calculated from known gas-phase chemistry (Sörgel et al., 2011a). Identification of the sources of
atmospheric HONO and exploration of its formation mechanisms are beneficial for enhancing our comprehension of
atmospheric chemistry processes and improving the accuracy of atmospheric simulation models.
Commonly accepted HONO sources include direct emission from motor vehicles (Chang et al., 2016; Kirchstetter et al.,
1996; Kramer et al., 2020; Xu et al., 2015) or soil (Su et al., 2011; Tang et al., 2019; Oswald et al., 2013), the homogeneous
conversion of NO by OH (Seinfeld and Pandis, 1998; Kleffmann, 2007), and the heterogeneous reaction of $NO_2$ on humid
surfaces (Alicke, 2002; Finlayson-Pitts et al., 2003). Other homogeneous sources include  nucleation reaction of $NH_3$, $NO_2$,
and $H_2O$ (Zhang and Tao, 2010), electronically excited $H_2O$ and $NO_2$ for the production of HONO (Li et al., 2008), the
$HO_2 \cdot H_2O$ complex and $NO_2$ for the production of HONO (Li et al., 2014). Other heterogeneous sources, include $NO_2$
reduced on soot to produce HONO and drastically enhanced by light(Ammann et al., 1998; Monge et al., 2010), semivolatile
organics from diesel exhaust for the production of HONO (Gutzwiller et al., 2002), photoactivated of $NO_2$ on humic acid
(Stemmler et al., 2006), $TiO_2$ (Ndour et al., 2008),solid organic compounds(George et al., 2005), the photolysis of particulate
nitrate by ultraviolet (UV) light (Kasibhatla et al., 2018; Romer et al., 2018; Ye et al., 2017; Scharko et al., 2014), -
dissolution of $NO_2$ catalyzed by anion on aqueous microdroplets (Yabushita et al., 2009), the process of acid displacement
(Vandenboer et al., 2014), the conversion of $NO_2$ to HONO on the ground(Wong et al., 2011), $NH_3$ enhancing the





heterogeneous reaction of $NO_2$ with $SO_2$ for the production of HONO (Ge et al., 2019), $NH_3$ promoting $NO_2$ dimers
hydrolysis for HONO production through stabilizing the state of product and reducing the reaction free energy barrier (Li et
al., 2018b; Xu et al., 2019), heterogeneous formation of HONO catalyzed by $CO_2$ (Xia et al., 2021). Heterogeneous
processes are the most poorly understood, yet are widely considered the main sources of HONO in previous studies. The
uptake coefficients of $NO_2$ conversion to HONO on surfaces (including aerosol, ground, buildings, and vegetation) vary
from $10^{-9}$ to $10^{-2}$ derived from different experiments (Ammann et al., 1998; Kirchner et al., 2000; Underwood et al., 2001;
Aubin and Abbatt, 2007; Zhou et al., 2015; Liu et al., 2014; Vandenboer et al., 2013). It is still a challenge to extrapolate
laboratory results to real surfaces. It is still under exploration to distinguish the key step to determine the $NO_2$ uptake, and we
are also not sure what role does radiation play in it. Absence of major HONO sources during the daytime is another active
ongoing research.
According to an analysis of 15 sets of field observations around the world (Elshorbany et al., 2012), the $HONO/NO_x$ ratio
(0.02) predicts well HONO concentrations under different atmospheric conditions. To avoid underestimation of HONO in
this study, an empirical parameterization was applied to estimating the HONO concentration, because the current
understanding of HONO formation mechanisms is incomplete. Field measurements of HONO and its precursor $NO_2$ at sites
with different aerosol load & composition, photolysis rate constants, and meteorological parameters are necessary to deepen
our knowledge of the HONO formation mechanisms. Such measurements have been carried out in coastal cities in China,
including Guangzhou (Qin et al., 2009), Hong Kong (Xu et al., 2015), and Shanghai (Cui et al., 2018), where the air
pollution is relatively severe during their research period. However, there has been a lack of research into HONO in coastal
cities with good air quality, low concentrations of $PM_{2.5}$, but strong sunlight and high humidity. Insufficient research on
coastal cities with good air quality has resulted in certain obstacles to assessing the photochemical processes in these areas.
Due to different emission-source intensities and ground surfaces, the atmospheric chemistry of HONO in the southeastern
coastal area of China is predicted to have different pollution characteristics from those found in other coastal cities.
Furthermore, HONO contributes to the atmospheric photochemistry differently depending on the season (Li et al., 2010).
Therefore, observations of atmospheric HONO across different seasons in the southeastern coastal area of China are urgently
needed.
Incoherent broadband cavity-enhanced absorption spectroscopy (IBBCEAS) was employed in this study to determine
HONO concentrations in the southeastern coastal city of Xiamen in August (summer), October (autumn), and December
(winter) 2018 and March (spring) 2019. In addition, a series of other relevant trace gases, meteorological parameters, and
photolysis rate constants were measured at the same time to provide supplementary information to reveal the HONO
formation mechanisms. The main purposes of this study were to (1) calculate the values of unknown HONO daytime sources,
(2) analyze the processes leading to HONO formation, (3) simulate HONO concentrations based on an empirical
parameterization, and (4) evaluate OH production from HONO from 07:00 to 16:00 local time. These results were compared
between the seasons.


## 2 Methodology

### 2.1 Site description

Our field observations were carried out ~80 m above the ground at a supersite located on the top of the Administrative Building of the Institute of Urban Environment (IUE), Chinese Academy of Sciences (118°04′13″E, 24°36′52″N) in Xiamen, China in August, October, and December 2018, and March 2019 (Fig. 1). The supersite was equipped with a complete set of measurement tools, including those for measuring gas and aerosol species composition, meteorology parameters, and photolysis rate constants, which provided a good chance to study the atmospheric chemistry of HONO in a coastal city of southeastern China. As shown in Fig. 1 (left), Xiamen is located at the southeast coastal area of China and faces the Taiwan Strait in the east. It suffers from sea and land breeze throughout the year with spring and summer more frequently (Xun et al., 2017). The IUE supersite is surrounded by a Xinglin Bay, several universities (or institutes), and several major roads with large traffic fleet, such as Jimei Road, Shenhai Expressway (870 m) and Xiasha Expressway (2300 m) (Fig. 1 (right)). The area of Xiamen is 1700.61 km$^2$ with a population of 4.11 million (http://tjj.xm.gov.cn/tjzl/). The number of motor vehicles in 2018 was 1,572,088, which was 2.73 times as many as ten years ago. The surrounding soil is used for green not for agriculture.

### 2.2 Instrumentation

The atmospheric concentrations of both HONO and NO$_2$ were determined using IBBCEAS, which has previously been widely applied to such measurements (Tang et al., 2019; Duan et al., 2018; Min et al., 2016). The IBBCEAS instrument was customized by the Anhui Institute of Optics and Fine Mechanic (AIOFM), Chinese Academy of Sciences (Duan et al., 2018). The resonant cavity is composed of a pair of high reflective mirrors separated by 70 cm and their reflectivity is approximately 0.99983 at 368.2 nm. The surface of the mirrors was purged by dry nitrogen at 0.1 Standard Liter per Minute (SLM), and the air flow was controlled by mass flow controller to prevent the surface of the mirror from being contaminated. Light was introduced into the resonant cavity and was emitted by a single light-emitting diode (LED) with full width at half maximum (FWHM) of 13 nm, peak wavelength of 365 nm. Light transmitted through the cavity was received by an QE65000 spectrometer (Ocean Optics) through an optical fiber with 600 μm diameter and a 0.22 numerical aperture.

In order to avoid the drift of the center wavelength of the LED, the temperature of the LED was controlled to be approximately 25 ± 0.01 °C by using a thermoelectric cooler unit. In order to prevent particulate matter from entering the cavity and reducing the effect of particulate matter on the effective absorption path, a 1 μm polytetrafluoroethylene (PTFE) filter membrane (Tisch Scientific) was used in the front end of the sampling port. In order to ensure the quality of the data, the 1 μm PTFE filter membrane was usually replaced once every three days and the sampling tube was thoroughly cleaned with alcohol once a month. We increased the replacement frequency of the filter membrane and the cleaning frequency of the sampling tube in the event of heavy pollution to ensure that the filter membrane and sampling tube are in a clean state.



The length of sampling tube with 6 mm outer diameter was approximately 3 m, the material was PFA with excellent
chemical inertness and the sampling flow rate was 6 SLM meaning that the residence time of the gas in the sampling tube
was less than 0.5 s. Besides, the sampling loss was calibrated before the experiment. We assessed the measured spectrum
every day to ensure the authenticity of the measurement results. Multiple reflections in the resonator cavity enhanced the
length of the effective absorption path, thereby enhancing the detection sensitivity of the instrument. The $1\sigma$ detection limits
for HONO and $NO_2$ were about 60 ppt and 100 ppt, respectively, and the time resolution was 1 min. The fitting wavelength
range was selected as 359–387 nm. The measurement error of HONO of IBBCEAS was estimated to be about 9 %,
considering both HONO secondary formation and sample loss. The sampling tube was heated to 35 °C and covered by
insulation cotton materials to prevent the effect of condensation of the water vapor(Lee et al., 2013).
The inorganic composition of $PM_{2.5}$ aerosols ($SO_4^{2-}$, $NO_3^-$, $Cl^-$, $Na^+$, $NH_4^+$, $K^+$, $Ca^{2+}$, and $Mg^{2+}$) and concentrations of gases
(HONO, $HNO_3$, HCl, $SO_2$, $NH_3$) were determined using a Monitor for AeRosols and Gases in ambient Air (MARGA, Model
ADI 2080, Applikon Analytical B.V., the Netherlands). Ambient air was drawn into the sample box by a $PM_{2.5}$ cyclone
(Teflon coated, URG-2000-30ENB) at the flow rate of 1 $m^3 \cdot h^{-1}$. Air sample was drawn firstly through the Wet Rotating
Denuder (WRD) where gases diffused to the solution, and then particles were collected by a Steam Jet Aerosol Collector
(SJAC). Absorption solutions were drawn from the SJAC and the WRD to syringes (25 ml).  Samples were injected to
Metrohm cation (500 µl loop) and anion (250 µl loop) chromatographs with the internal standard (LiBr) for 15 min after an
hour when the syringes had been filled (Makkonen et al., 2012). Specific descriptions of the SJAC can be found in previous
reports (Slanina et al., 2001; Wyers et al., 1993). Therefore, the times needed for the sampling period and the latter IC
analysis on the MARGA system are a full hour and 15 minutes, respectively. The value measured in this hour is actual the
concentration sampled in the previous hour, so the time corresponding to the sampling is matched with other instrument
parameters (i.e., HONO, NO$x$, $J$ values).
Photolysis frequencies were determined using a photolysis spectrometer (PFS-100, Focused Photonics Inc., Hangzhou,
China). These were calculated by multiplying the actinic flux $F$, quantum yield $\varphi(\lambda)$ and the known absorption cross section
$\sigma(\varphi)$. The measurements included the photolysis rate constants $J$ ($O^1D$), $J$ (HCHO_M), $J$ (HCHO_R), $J$ ($NO_2$), $J$ ($H_2O_2$), $J$
(HONO), $J$ ($NO_3$_M) and $J$ ($NO_3$_R), and the spectral band ranged from 270 to 790 nm. Hemispherical ($2\pi$ sr) angular
response deviations were within $\pm 5$ %. The photolysis rate constants with _R and _M represented radical photolysis channel
and molecular photolysis channel, respectively. Specifically, HCHO was removed by the reactions (R1) and (R2), and $NO_3$
was removed by the reactions R(3) and R(4), respectively (Röckmann et al., 2010).
$HCHO + h\nu \longrightarrow CHO + H$     $J(HCHO\_R)$        (R1)
$HCHO + h\nu \longrightarrow H_2 + CO$     $J(HCHO\_M)$        (R2)
$NO_3 + h\nu \longrightarrow NO_2 + O^3P$     $J(NO_3\_R)$        (R3)



$NO_3 + hv \longrightarrow NO + O_2$      $J(NO_3\_M)$      (R4)
The $O_3$ concentration was determined by UV photometric analysis [Model 49$i$, Thermo Environmental Instruments (TEI)
Inc.], and the detection limit of the TEI Model 49$i$ is 1.0 ppb. The NO concentration was determined by a
chemiluminescence analyzer (TEI model 42$i$) with a molybdenum converter. The detection limit and the uncertainty of the
TEI model 42$i$ were 0.5 ppb and 10 %, respectively. Although the TEI model 42$i$ also measures the concentration of $NO_2$,
this value might actually include other active nitrogen components(Villena et al., 2012). As expected, the $NO_2$ concentration
measured by IBBCEAS had the same trend as the $NO_2$ measured by TEI 42$i$, and $NO_2$ concentration measured by IBBCEAS
was always lower than that by TEI 42i (Fig. S1). Therefore, the $NO_2$ concentration as measured by IBBCEAS was used in
this study. An oscillating microbalance with a tapered element was applied to determine the $PM_{2.5}$ concentration with
uncertainty of 10-20 %. Black carbon (BC) was measured by aethalometer at 7 wavelengths (in using 880 nm wavelength).
When the tape was < 10 %, aethalometer fiber tape was replaced. Meteorological parameters were determined by an
ultrasonic atmospherium (150WX, Airmar, USA). The time resolution of all instruments was unified to 1 h to facilitate
comparison. Ultraviolet radiation (UV) was determined by a UV radiometer (KIPP & ZONEN, SUV5 Smart UV
Radiometer).
**3 Results and discussion**
**3.1 Overview of data**
Figure 2 showed an overview of the determined HONO, NO, $NO_2$, $PM_{2.5}$, $NO_3^-$, BC, $J$(HONO), temperature (T) and relative
humidity (RH)in this study. The entire campaign was characterized by subtropical monsoon climate with high temperature
(9.82$-$34.42 °C) and high humidity (29.24$-$100 %). The mean values ($\pm$ standard deviation) of temperature and relative
humidity were 22.24 $\pm$ 5.41 °C and 78.35 $\pm$ 14.07 %, respectively. Elevated concentrations of NOx, i.e., up to 156.17 ppb of
NO, and 172.42 ppb of $NO_2$, were observed, possibly due to dense vehicle emissions near this site. The photolysis rate
constants $J(O^1D)$, $J$(HCHO_M), $J$(HCHO_R), $J(NO_2)$, $J(H_2O_2)$, $J$(HONO), $J(NO_3\_M)$ and $J(NO_3\_R)$ had the same temporal
variation (Fig. S2), although their orders of magnitude were different. The correlation coefficients between $J$(HONO) and
other photolysis rate constants were above 0.965 (not shown). Both $J$(HONO) and UV peaked around noon, and the
maximum of $J$(HONO) ($2.02 \times 10^{-3}$ $s^{-1}$) and UV (55.62 W·$m^{-2}$) appeared at 13:00 on 11 March 2019, and 12:00 on 14 August
2018, respectively. This area was dominated by photochemical pollution, while particulate pollution was relatively light., No
haze episodes occurred across four seasons with 111 days because daily mass concentration of $PM_{2.5}$ was lower than the
National Ambient Air Quality Standard (Class II: 75 $\mu g·m^{-3}$). For $O_3$, 10 episodes occurred with eight-hour maximum
concentrations of $O_3$ exceeding the Class II: 160 $\mu g·m^{-3}$. Maximum mixing ratio of $O_3$ was 113.81 ppb, occurring in the
afternoon with strong ultraviolet radiation (42.72 w·$m^{-2}$) and low NO concentration (0.75 ppb) titrating $O_3$. In general, the
level of pollutants in this area was relatively low. Campaign-averaged levels of $NO_2$, NO, $NO_3^-$, $PM_{2.5}$, $O_3$, and BC were



14.99 ± 8.93 ppb, 5.80 ± 11.98 ppb, 5.59 ± 6.26 $\mu g \cdot m^{-3}$, 27.78 ± 17.95 $\mu g \cdot m^{-3}$, 28.29 ± 21.14 ppb, and 1.67 ± 0.97 $\mu g \cdot m^{-3}$,
respectively.  The maximum value of HONO (3.51 ppb) appeared at 08:00 on 4 December 2018. The high value of HONO
was always accompanied by relative high values of NO and $NO_2$ or $PM_{2.5}$, BC and $NO_3^-$. The average measured ambient
HONO concentration at the measurement site for all measurement periods was 0.54 ± 0.47 ppb. The HONO concentration
measured at this site was comparable to those measured at other suburban sites (Liu et al., 2019c; Xu et al., 2015; Nie et al.,
2015a; Park et al., 2004), was obvious lower than those measured at urban sites and industrial site (Li et al., 2018a; Yu et al.,
2009; Hou et al., 2016; Qin et al., 2009; Wang et al., 2013; Shi et al., 2020; Spataro et al., 2013; Huang et al., 2017; Wang et
al., 2017), and was obvious higher than those measured at marine background(Wen et al., 2019a), Marine boundary layer(Ye
et al., 2016), and coastal remote (Meusel et al., 2016) , as shown in Table S1.
As shown in Table 1, in the daytime (06:00–18:00, including 06:00, local time (LT)), the highest concentration of HONO
was found in spring and summer (0.72 ppb), followed by winter (0.61 ppb) and autumn (0.50 ppb). In short, the seasonal
variation of HONO was well correlated with the seasonality of RH, with high RH in spring (84.21 %) and summer (84.12 %),
followed by winter (78.13 %) and autumn (69.55 %). In conditions of low RH, the adsorption rate of $NO_2$ is not as rapid as
that of HONO, resulting in a reduction in the conversion rate of $NO_2$ to HONO and thus a reduction in the concentration of
HONO (Stutz et al., 2004). This seasonal variation in HONO concentration was different from those measured in Jinan (Li et
al., 2018a), Nanjing (Liu et al., 2019b), and Hong Kong (Xu et al., 2015). The elevated HONO concentrations in summer,
when there is strong solar radiation, suggests the existence of strong sources of HONO and its important contribution to the
production of OH radicals. Interestingly, the HONO concentration in the nighttime was lower than that in the daytime in all
four seasons. Most previous studies have found that the HONO concentration at night is significantly higher than that during
the day (Wang et al., 2015; Liu et al., 2019c; Li et al., 2018a; Elshorbany et al., 2009; Acker et al., 2006; Yu et al., 2009).
Coastal cities are susceptible to sea and land breezes, with sea breezes blowing during the day and land breezes blowing
during the night (Wagner et al., 2012). Therefore, the concentration of sea salt, as calculated based a previous report (Liu et
al., 2020), is significantly higher during the day (2.91 $\mu g \cdot m^{-3}$) than that during the night (2.73 $\mu g \cdot m^{-3}$) ($P < 0.05$). It is
possible that significantly more HONO could be produced by photolysis of sea salts against the daytime photolysis of
HONO (Kasibhatla et al., 2018). Similar results were found in Hong Kong, which is also a coastal city, which further
validates the rationality of this assumption (Xu et al., 2015). As shown in Fig. 3, larger difference between daytime and
nighttime HONO concentrations was observed on days with SLBs compared without SLBs, which indicated that SLBs did
cause higher HONO concentration in the daytime than that in the nighttime.
The ratio of HONO to $NO_x$ or the ratio of HONO to $NO_2$ have been extensively applied to indicate heterogeneous conversion
of $NO_2$ to HONO (Li et al., 2012b; Liu et al., 2019c; Zheng et al., 2020). Compared with the $HONO/NO_2$ ratio, the
$HONO/NO_x$ ratio can better avoid the influence of primary emissions (Liu et al., 2019c). In this study, the $HONO/NO_x$ ratios
during the day were higher than those during the night, indicating that light promotes the conversion of $NO_x$ to HONO. The
highest daytime $HONO/NO_x$ ratio was found in summer (0.072), followed in turn by autumn (0.048), spring (0.034), and



winter (0.023). The elevated HONO/NO$_x$ ratio in summer indicates a greater net HONO production (Xu et al., 2015). The
low HONO/NO$_x$ ratio in winter can probably be ascribed to heavy emissions and high concentrations of NO in winter
(Table 1). The HONO/NO$_x$ ratios during every season in Xiamen were in general higher than those found in studies of other
cities, which indicates greater net HONO production in Xiamen.
The diurnal patterns of HONO, NO$_x$, HONO/NO$_x$, and $J$(NO$_2$) averaged for every hour in each season are shown in Fig. 4.
As shown in Fig. 4a, the HONO concentration had similar diurnal variation patterns across the four seasons. The maximum
values of the HONO concentration were 1.12 ppb in winter, 1.03 ppb in summer, 0.98 ppb in spring, and 0.65 ppb in autumn,
and these occurred in the morning rush hour (07:00–08:00), which indicates that direct vehicle emissions may be a
significant source of HONO. The contribution of direct vehicle emissions to HONO will be quantified in Sect. 3.2. The
HONO concentration reduced rapidly from the morning rush hour to sunset, and this was caused by rapid photolysis
combined with increased height of the boundary layer. The minimum values of HONO concentration were 0.47 ppb in
spring, 0.23 ppb in winter, 0.21 ppb in summer, and 0.14 ppb in autumn, and these appeared at sunset, between 16:00 and
18:00. The HONO concentration increased gradually after sunset, which indicates that release from HONO sources exceeded
its dry deposition (Wang et al., 2017). There was a slight difference in the diurnal variation of HONO between autumn and
the other seasons. A rapid reduction of HONO after the morning rush hour was found in spring, summer, and winter. In
comparison, the HONO in autumn had an almost constant concentration between 07:00 and 11:00 because NO$_x$ decreased
slowly during this period.
As shown in Fig. 4b, NO$_x$ concentration followed an expected profile in the four seasons, with peaks of 45.58 ppb in winter,
40.47 ppb in spring, 32.47 ppb in summer, and 20.07 ppb in autumn, each occurring in the morning rush hour at 10:00, 09:00,
08:00, and 07:00 local time, respectively. After these peaks, NO$_x$ decreased during the day in each season, probably due to
photochemical transformation and increasing boundary-layer depth. The NO$_x$ concentrations then began to rise from their
minima of 8.20 ppb in summer, 8.85 ppb in autumn, 18.10 ppb in winter, and 23.09 ppb in spring after 14:00, 13:00, 15:00,
and 16:00 local time, respectively, which was caused by a combination of weak photochemical transformation and reduction
in the boundary-layer depth. The NO$_x$ concentrations during winter and spring were significantly higher than those during
autumn and summer. Both the maxima and minima of NO$_x$ appeared later in spring and winter compared with summer and
autumn.
It is possible to better describe the behavior of HONO using the HONO/NO$_x$ ratio. The higher HONO/NO$_x$ ratio found at
noon in the different seasons, especially in summer and autumn (Fig. 4c), indicates an additional daytime HONO source(Liu
et al., 2019c; Xu et al., 2015). It is worth noting that the maximum value of this ratio in summer (0.147) was significantly
higher than the maximum in other seasons, especially in winter (0.034). Fig. 4d shows that the value of the HONO/NO$_x$ ratio
increased with the photolysis of NO$_2$ in summer and autumn, suggesting that the additional HONO source is probably
correlated with light (Xu et al., 2015; Wang et al., 2017; Li et al., 2018a; Li et al., 2012b). The increase in the HONO/NO$_2$
ratio during the day can be seen more clearly in Fig. 5, and its high value indicates a high HONO production efficiency,



which cannot be ascribed to $NO_2$ conversion due to the weak correspondence between HONO and $NO_2$ in in summer.
Furthermore, high HONO/$NO_2$ ratios were accompanied by high $J(NO_2)$ in summer, which indicates that HONO formation
during the daytime is controlled by light rather than Reaction (R5).
$$NO_2 + NO_2 + H_2O \xrightarrow{surf} HONO + HNO_3 \qquad\qquad\qquad (R5)$$
However, the observed maxima can also be ascribed to sources independent from $NO_x$ concentration, such as soil emissions
(Su et al., 2011) and photolysis of particulate nitrate (Zhou et al., 2011; Ye et al., 2016), which are not influenced by the
decrease of $NO_x$ concentration around noon. A more specific discussion of daytime HONO sources considering the
photolysis of particulate nitrate will be given in Sect. 3.4.3. The HONO emissions from soil were estimated to be 2–5 ppb h$^{-1}$
(Su et al., 2011). However, soil emission was a negligible source of HONO in this study since the surrounding soil is not
used for agriculture, and this greatly reduces the amount of HONO released due to no fertilization process (Su et al., 2011).

### 3.2 Direct vehicle emission of HONO

The K$^+$ levels were 0.26, 0.13, 0.14, and 0.24 µg·m$^{-3}$ for spring, summer, autumn, and winter, respectively. The K$^+$ levels
during the four seasons were lower than 2 µg·m$^{-3}$, which indicated that biomass burning has little effect on this site (Nie et al.,
2015b; Xu et al., 2019). Hence, only vehicle emissions were considered in this study. The consistent diurnal variations in
HONO and $NO_x$ presented in Sect. 3.1 (Fig. 4) also indicate HONO emissions from local traffic. Five criteria were applied to
choose cases that guaranteed the presence of fresh plumes (Xu et al., 2015; Liu et al., 2019c): (1) UV < 10 W·m$^{-2}$; (2) short-
duration air masses (<2 h); (3) HONO correlating well with $NO_x$ ($R^2$ > 0.60, $P$ < 0.05); (4) $NO_x$ > 20 ppb (highest 25 % of
$NO_x$ value); and (5) NO/$NO_x$ > 0.50. A total of 34 cases met these strict criteria for estimation of the HONO vehicle
emission ratios. The slopes of scatter plots of HONO vs $NO_x$ were used as the emission factors.
A total of 34 vehicle emission plumes were summarized in Table 2, and these were used for estimation of the vehicle
emission ratios. These plumes were considered to be truly fresh because the mean $\Delta NO/\Delta NO_x$ ratio of the selected air masses
was 92 %. Vehicle plumes unavoidably mixing with other air masses resulted in the correlation coefficients ($R^2$) between
HONO and $NO_x$ varying among the cases, and these ranged from 0.61to 0.92. The obtained $\Delta HONO/\Delta NO_x$ ratios ranged
from 0.24 % to 4.76 %, with an average value (±SD) of (1.64 ± 0.95) %. These $\Delta HONO/\Delta NO_x$ ratios have comparability to
those obtained in Guangzhou (1.4 % (Qin et al., 2009); 1.8 % (Li et al., 2012b)) and Houston (1.7 % (Rappenglück et al.,
2013)), but are significantly higher than those measured in Jinan (0.53 % (Li et al., 2018a)) and Santiago (0.8 % (Elshorbany
et al., 2009)). The types of vehicle engine, the use of catalytic converters, and different fuels will affect the vehicle emission
factors (Kurtenbacha et al., 2001). A potential reason for the relatively higher $\Delta HONO/\Delta NO_x$ values in our study is that
heavy-duty diesel vehicles pass by on the surrounding highway (Rappenglück et al., 2013). It is necessary to examine the
specific vehicle emission factors in target cities because of these differences in $\Delta HONO/\Delta NO_x$ ratios. Roughly assuming that





$NO_x$ mainly arises from vehicle emissions, a mean $\Delta HONO/\Delta NO_x$ value of 1.64 % was used as the emission factor in this
study, and this value was adopted to estimate the contribution of vehicle emissions $P_{emis}$ to the HONO concentration using
$$P_{emis} = NO_x \times 0.0164. \tag{1}$$
We can then obtain the corrected HONO concentration ($HONO_{corr}$) for further analysis from the equation
$$HONO_{corr} = HONO - P_{emis}. \tag{2}$$

### 289    3.3 Nighttime heterogeneous conversion of NO₂ to HONO

### 290    3.3.1 Conversion rate of NO₂ to HONO

Nighttime $HONO_{corr}$ concentrations can be estimated from the heterogeneous conversion reaction (Meusel et al., 2016;
Alicke, 2002; Su et al., 2008c). Although the mechanism of the nighttime HONO heterogeneous reaction is unclear, the
formula for the heterogeneous conversion ($C_{HONO}^0$) of NO₂ to HONO can be expressed as
$$C_{HONO}^0 = \frac{[HONO_{corr}]_{t_2} - [HONO_{corr}]_{t_1}}{(t_2 - t_1) \times \overline{[NO_2]}}, \tag{3}$$
where $\overline{[NO_2]}$ is the mean value of NO₂ concentration between $t_1$ and $t_2$. Eq. (4) has been suggested as a way to avoid the
interference of direct emissions and diffusion (Su et al., 2008c):
$$C_{HONO}^X = \frac{\left(\frac{[HONO_{corr}]_{(t_2)}}{[X]_{t_2}} - \frac{[HONO_{corr}]_{(t_1)}}{[X]_{(t_1)}}\right)\overline{[X]}}{(t_2 - t_1)\frac{1}{2}\left(\frac{[NO_2]_{(t_2)}}{[X]_{(t_2)}} + \frac{[NO_2]_{(t_1)}}{[X]_{(t_1)}}\right)\overline{[X]}} = \frac{2\left(\frac{[HONO_{corr}]_{(t_2)}}{[X]_{t_2}} - \frac{[HONO_{corr}]_{(t_1)}}{[X]_{(t_1)}}\right)}{(t_2 - t_1)\left(\frac{[NO_2]_{(t_2)}}{[X]_{(t_2)}} + \frac{[NO_2]_{(t_1)}}{[X]_{(t_1)}}\right)}, \tag{4}$$
where $[HONO_{corr}]_t$, $[NO_2]_t$, and $[X]_t$ were the concentrations of HONO, NO₂, and species used for normalization (including
NO₂, CO, and black carbon (BC) in this study), respectively, at time $t$, $\overline{X}$ is the average concentration of reference species
between $t_1$ and $t_2$, and $C_{HONO}^X$ represents the conversion rate normalized against reference species $X$ (Su et al., 2008c). There
were 91 cases meeting the criteria. Such a large number of cases contributes to the statistical analysis of the heterogeneity of
HONO formation. The average values of $C_{HONO}^0$, $C_{HONO}^{NO_2}$, $C_{HONO}^{CO}$, and $C_{HONO}^{BC}$ were 0.48 % h⁻¹, 0.46 % h⁻¹, 0.47 % h⁻¹, and 0.46
% h⁻¹, respectively. The combined $C_{HONO}^C$ was 0.47 % h⁻¹. The average $C_{HONO}$ values obtained using different normalization
methods agreed well. Therefore, an estimation value of 0.47 % h⁻¹ should be suitable for the nighttime conversion rate from
NO₂ to HONO.
We also compared the conversion rates calculated in this study with other experiments. As shown in Table 3, $C_{HONO}^C$ varied
widely, from 0.29 % h⁻¹ to 2.40 % h⁻¹, which may be due to the various kinds of land surface in the various environments.
The $C_{HONO}^C$ in Xiamen is comparable to those derived in Shanghai (0.70 % h⁻¹ (Wang et al., 2013)), Jinan (0.68 % h⁻¹ (Li et
al., 2018a)), and Hong Kong (0.52 % h⁻¹ (Xu et al., 2015)), less than the values calculated from most other sites, including



Xinken (1.60 % h$^{-1}$ (Su et al., 2008c)), Guangzhou (2.40 (Li et al., 2012b)), Spain (1.50 (Sörgel et al., 2011a)), Beijing (0.80
(Wang et al., 2017)), the eastern Bohai Sea (1.80 % h$^{-1}$ (Wen et al., 2019a)), and Kathmandu (1.40 % h$^{-1}$ (Yu et al., 2009)),
but more than the value obtained in Shandong (0.29 % h$^{-1}$ (Wang et al., 2015)). The highest $C^{C}_{HONO}$ was found in summer,
with a value of 0.55 % h$^{-1}$, which will be explained in Sect. 3.3.2. Another study also found that the highest $C^{C}_{HONO}$ (1.00
% h$^{-1}$) appeared in summer (Wang et al., 2017).
**3.3.2 The influence factors on HONO formation**
The hydrolysis of NO$_2$ on wet surfaces producing HONO is first-order affected by the concentration of NO$_2$ (Finlayson-Pitts
et al., 2003; Jenkin et al., 1988) and the absorption of water on the surfaces (Finlayson-Pitts et al., 2003; Kleffmann et al.,
1998). A scatter plot of HONO$_{corr}$/NO$_2$ vs RH is shown in Fig. 6. We calculated the top-five HONO$_{corr}$/NO$_2$ ratios in every 5
% RH interval based on a method introduced in previous literature (Li et al., 2012b; Stutz et al., 2004), which will reduce the
influence of those circumstances such as advection, the time of the night, and the surface density. These averaged maxima
and standard deviations are shown in Fig. 6 as orange squares, except where data were sparse in a particular 5 % RH interval.
As for autumn and winter, the influence of RH on HONO$_{corr}$/NO$_2$ can be divided into two parts. The RH promoted an
increase in HONO$_{corr}$/NO$_2$ for RH values less than 77.96 % in autumn and 91.99 % in winter, which is in line with the
reaction kinetics of Reaction (R5). However, RH inhibits the conversion of NO$_2$ to HONO when RH is higher than a turning
point. According to many previous studies, water droplets will be formed on the surface of the ground or of aerosols when
RH exceeds a certain value, thus resulting in a negative dependence of HONO$_{corr}$/NO$_2$ on RH (He et al., 2006; Zhou et al.,
2007). A similar phenomenon was also found in Guangzhou and in Shanghai (70 %, (Li et al., 2012b; Wang et al., 2013))
and in Kathmandu and in Beijing (65 %, (Yu et al., 2009; Wang et al., 2017)). However, in summer, RH appeared to
promote the increase of HONO$_{corr}$/NO$_2$ without a turning point, suggesting that HONO production at night in summer
strongly depends on RH. Another study also found a similar phenomenon in the summer in Guangzhou (Qin et al., 2009).
This phenomenon might be caused by water droplets being evaporated by high temperatures. This is the reason for the
highest $C^{C}_{HONO}$ in summer. As for spring, the relationship between HONO$_{corr}$/NO$_2$ and RH is very complicated and needs to
be explored further in the future.
It has been found that NH$_3$ promoted hydrolysis of NO$_2$ and production of HONO and NH$_4$NO$_3$(Xu et al., 2019; Li et al.,
2018b). The correlations between the HONO$_{corr}$/NO$_2$ ratio, the NO$_3^-$/NO$_2$ ratio and the NH$_3$ concentration in four seasons
were examined to investigate the influence of NH$_3$ on HONO formation through promoting hydrolysis of NO$_2$. Only
nighttime data with RH above 80 % were chosen to avoid daytime rapid photolysis of HONO and enough water for NO$_2$
quick hydrolysis.(Xu et al., 2019). As shown in Fig. 7, for summer, the correlations between NH$_3$ and HONO$_{corr}$/NO$_2$ ratio
was very poor and even negative (R=-0.0396), and the correlation between NO$_3^-$/NO$_2$ ratio and NH$_3$ was also negative (-
0.2741). These results indicated that NH$_3$ played a minor role in HONO production in summer. For autumn, although NO$_3^-$
/NO$_2$ ratio correlated well with NH$_3$ (R=0.3431) in autumn, HONO$_{corr}$/NO$_2$ ratio had bad correlation with NH$_3$ (R=0.0843),


which also indicated that $NH_3$ played a minor role in HONO production in autumn. For spring, the correlation coefficient
between the $HONO_{corr}/NO_2$ ratio and the $NH_3$ concentration was highest among four seasons (0.3664), and the correlation
between the $NO_3^-/NO_2$ ratio and the $NH_3$ concentration was positive (0.1452). These phenomena proved that $NH_3$ might
promote HONO and $NH_4NO_3$ production through promoting $NO_2$ hydrolysis in spring. For winter, medium correlations were
found in $NH_3$ with both $HONO/NO_2$ ratio (R=0.2131) and $NO_3^-/NO_2$ ratio (R=0.2556), which indicated that $NH_3$ might
promote $NO_2$ hydrolysis and HONO production in winter. All in all, $NH_3$ might promote $NO_2$ hydrolysis and HONO
production in spring and winter, whereas $NH_3$ played a minor role in HONO production in summer and autumn.
As shown in Fig. S3, $HONO_{corr}/NO_2$ reached a pseudo-steady state from 03:00 to 06:00 LT every night. A correlation
analysis of $HONO_{corr}/NO_2$ with $PM_{2.5}$ was carried out in the pseudo-steady state to understand the impact of aerosols on
HONO production. Although we did not measure the aerosol surface density, the aerosol mass concentration can be used to
replace this parameter (Huang et al., 2017; Park et al., 2004; Cui et al., 2018). The positive correlation of $HONO_{corr}$ with
$PM_{2.5}$ ($R_1 = 0.54$) (Fig. 8a) may be a result of atmospheric physical processes such as convergence and diffusion. Using the
$HONO_{corr}/NO_2$ ratio instead of a single HONO concentration for correlation analysis with $PM_{2.5}$ reduce the impact of
physical processes and indicate the extent of conversion of $NO_2$ to HONO. Therefore, it was more credible that
$HONO_{corr}/NO_2$ would be moderately positively correlated with $PM_{2.5}$ ($R_2 = 0.23$) during the whole observation period
(Fig. 8b). As denoted by larger green squares in the Fig. 8b, $HONO_{corr}/NO_2$ correlated well with $PM_{2.5}$ when its
concentration was higher than 35 $\mu g \cdot m^{-3}$ ($R_3 = 0.47$). The larger the amount of HONO produced by the heterogeneous
reaction of $NO_2$ on the aerosol surface, the better the correlation between $HONO/NO_2$ and $PM_{2.5}$ (Cui et al., 2018; Wang,
2003; Hou et al., 2016; Li et al., 2012b; Nie et al., 2015a).

### 3.4 Daytime sources of HONO

### 3.4.1 Budget analysis of HONO

Having discussed the nighttime chemical behavior of HONO, we now concentrate on the daytime chemical behavior of
HONO. Here, $R_{unknown}$ is used to stand for the rate of emission from unknown sources. The value of $R_{unknown}$ was estimated
based on the balance between sources and sinks due to its short atmospheric lifetime. The sources are: (1) oxidation of NO
by OH ($R_{OH+NO} = k_{OH+NO}[NO][OH]$), (2) dark heterogeneous production ($P_{het}$), and (3) direct vehicle emission ($P_{emis}$); the
sinks are (1) HONO photolysis ($R_{phot} = J_{HONO}[HONO]$), (2) oxidation of HONO by OH ($R_{OH+HONO} =$
$k_{OH+HONO}[HONO][OH]$), and (3) dry deposition ($L_{dep}$). The value of $R_{unknown}$ can then be calculated according to
$$R_{unknown} = J_{HONO}[HONO] + k_{OH+HONO}[HONO][OH] + L_{dep} + \frac{\Delta[HONO]}{\Delta t} - k_{OH+NO}[NO][OH] - P_{het} - P_{emis}, \qquad (5)$$
Where $k_{OH+HONO} = 6.0 \times 10^{-12}$ $cm^3$ molecules$^{-1}$ s$^{-1}$ and $k_{OH+NO} = 7.4 \times 10^{-12}$ $cm^3$ molecules$^{-1}$ s$^{-1}$, values cited from a previous
study(Sörgel et al., 2011b). The OH concentration ([OH]) was estimated in this study because no data for this value were
available. An improved empirical formula, Eq. (6), was applied to estimate [OH] using the $NO_2$ and HONO concentrations





and the photolysis rate constants ($J$) of $NO_2$, $O_3$, and HONO(Wen et al., 2019b). Eq. (6) fully considers the influence of
photolysis and precursors on the concentration of [OH].
$$[OH] = 4.1 \times 10^9 \times \frac{J(O^1D)^{0.83} \times J(NO_2)^{0.19} \times (140 \times NO_2 + 1) + HONO \times J(HONO)}{0.41 \times NO_2^2 + 1.7 \times NO_2 + 1 + NO \times k_{NO+OH} + HONO \times k_{HONO+OH}}$$ (6)
During spring, summer, autumn, and winter, the average midday OH concentrations were $8.86 \times 10^6$ cm$^{-3}$, $1.48 \times 10^7$ cm$^{-3}$,
$1.36 \times 10^7$ cm$^{-3}$, and $6.19 \times 10^6$ cm$^{-3}$, respectively, which were within the range of those obtained in other studies varying
from $4 \times 10^6$ cm$^{-3}$ to $1.7 \times 10^7$ cm$^{-3}$ (Tan et al., 2017; Lu et al., 2013).
$\frac{\Delta[HONO]}{\Delta t}$ is the observed change of HONO concentration(ppb·s$^{-1}$). The value of $\frac{\Delta[HONO]}{\Delta t}$ is the concentration difference
between the center of one interval (1 min) and the center of the next interval, and this accounts for changes in concentration
levels (Sörgel et al., 2011a). The parameter $L_{dep}$ can be quantified by multiplying the dry deposition rate of HONO by the
observed HONO concentration and then dividing by the mixing layer height ( $L_{dep} = \frac{v_{HONO}^{ground} \times [HONO]}{H}$ ). A value of
$v_{HONO}^{ground} = 2$ cm·s$^{-1}$ was used for the deposition rate (Sörgel et al., 2011a; Su et al., 2008b). The mixing layer heights during
spring, summer, autumn, and winter were 1074.4 m, 1173.8 m, 1494.6 m, and 1310.4 m, respectively (Gao, 1999). In
summarizing the known HONO sources, we included the nighttime heterogeneous production as a known source based on
the assumption that the day continues in the same way as the night (Sörgel et al., 2011a). The term $P_{het}$ was parameterized by
$NO_2$ conversion at night using the formula $P_{het} = C_{HONO}^{C}[NO_2]$ (Alicke, 2002).
Figure 9 shows the contributions of each term in Eq. (7) to the HONO budgets in different seasons. Photolysis of HONO
($R_{phot}$) formed the largest proportion of the sinks in all four seasons, accounting for 95.33 %, 94.60 %, 95.46 %, and 96.18 %
in spring, summer, autumn, and winter, respectively. The value of $R_{phot}$ in summer was the highest (3.65 ppb·h$^{-1}$), followed
by spring (3.17 ppb·h$^{-1}$), winter (2.48 ppb·h$^{-1}$) and autumn (2.42 ppb·h$^{-1}$). The oxidation of HONO by OH contributed little
to HONO sinks (2.97 % of all sinks). Dry deposition ($L_{dep}$) was also very small (1.63 % of all sinks). As for known sources,
$R_{OH+NO}$ was the main known source in all four seasons, wherein the largest proportion was found in summer (63.24 %),
followed by autumn (52.36 %), spring (52.02 %), and winter (49.99 %). Direct emission was second among the known
sources, accounting for 39.58 %, 29.15 %, 38.36 %, and 43.03 % in spring, summer, autumn, and winter, respectively. Dark
heterogeneous formation ($P_{het}$) was almost negligible in the daytime, accounting for approximately 8.07 % of known sources
during the whole observation period. As for unknown sources, these made up the largest proportion of all sources found in
summer (79.55 %), followed by autumn (71.51 %), spring (69.67 %) and winter (55.63 %).
It is worth noting that $R_{unknown}$ exhibited a maximum around noon in all seasons. A previous study in Wangdu (Liu et al.,
2019d) also found that unknown sources of HONO reached a maximum at midday, with the strongest photolysis rates in
summer. This strengthens the validity of the assumption that the missing HONO formation mechanism is related to a
photolytic source (Michoud et al., 2014). In the present study, the daily maximum $R_{unknown}$ value was 4.35 ppb·h$^{-1}$ in summer,





followed by 3.53 ppb·h⁻¹ in spring, 3.13 ppb·h⁻¹ in autumn and 2.05 ppb·h⁻¹ in winter. Average $R_{unknown}$ during the whole
observation was 2.16 ppb·h⁻¹, which was almost at the upper-middle level of studies reported: 0.5 ppb·h⁻¹ in a forest near
Jülich, Germany(Kleffmann, 2005); 0.77 ppb·h⁻¹ at a rural site in the Pearl River delta, China (Li et al., 2012a); 1.04 ppb·h⁻¹
at a suburban site in Nanjing, China(Liu et al., 2019a); ≈2 ppb·h⁻¹ in Xinken, China(Su et al., 2008a); and 2.95 ppb·h⁻¹ in the
urban atmosphere of Jinan, China(Li et al., 2018a).

### 408   3.4.2 Exploration of possible unknown daytime sources

According to the analyses in Sect. 3.1 and Sect. 3.4.1, the unknown sources are likely to be related to light. It was indeed
found that the unknown sources have a good correlation with the parameters related to light. It was reported in previous
studies that particulate nitrate photolysis is a source of HONO (Ye et al., 2017; Ye et al., 2016; Scharko et al., 2014; Romer
et al., 2018; Mcfall et al., 2018). We will discuss the possibility of HONO being produced by photolysis of particulate nitrate
($J(NO_3\_R) \times pNO_3^-$) at this site in the next section. There was a logarithmic relationship showing good correlation between
$R_{unknown}$ (ppb·h⁻¹) and $J(NO_3^-\_R) \times pNO_3^-$ (µg·m⁻³·s⁻¹) in spring ($R^2 = 0.5982$) and summer ($R^2 = 0.5837$), while relatively
weak correlation was found in autumn ($R^2 = 0.2131$) and winter ($R^2 = 0.3764$) (Fig. 10). This result indicated that photolysis
of particulate nitrate contributed more in spring and summer than in autumn and winter. In conditions of relatively lower
$J(NO_3\_R) \times pNO_3^-$, $R_{unknown}$ increased rapidly with increasing $pNO_3^-$ concentration and its photolysis rate constant but
reached a plateau after a critical value ($J(NO_3\_R) \times pNO_3^- > 0.5$ µg·m⁻³·s⁻¹ in summer and autumn, and $J(NO_3\_R) \times$
$pNO_3^- > 1.5$ µg·m⁻³·s⁻¹ in winter). There was no obvious turning point in spring, but it could be seen that the growth rate
was declining. This indicated that in conditions that were relatively cleaner, the missing daytime source of HONO was
limited by the $pNO_3^-$ concentration and the photolysis rate constant. However, with enough particulate nitrate providing
sufficient precursor or enough light to stimulate the reaction, the HONO production did not increase as $J(NO_3\_R) \times pNO_3^-$
increased. Other generation mechanisms might play leading roles in the condition with enough particulate nitrate or enough
light. It was found in a previous study that heterogeneous soot photochemistry may contribute to the daytime HONO
concentration(Monge et al., 2010). Black carbon (BC) values were as a substitute for soot values(Sörgel et al., 2011b). When
BC concentration was above 2.0 µg·m⁻³, the missing daytime source of HONO did not increase as $J(NO_3\_R) \times pNO_3^-$
increased. We found that the missing daytime source of HONO correlated better with BC×UV (R=0.9247, R=0.6421) than
with BC (R=0.5012, R=0.5720) or UV (R=0.8556, R=0.4230) alone in autumn and winter (Fig. S4), probably related to the
conversion of NO₂ to HONO on BC enhanced by light.
We discuss whether photolysis of particulate nitrate was able to provide enough additional HONO by estimating the rate of
HONO production by nitrate photolysis in spring and summer (Zhou et al., 2007; Li et al., 2012b; Wang et al., 2017) using
$$J_{NO_3^- \rightarrow HONO} = \frac{R_{unknown} \times H}{f \times [NO_3^-] \times \upsilon_{NO_3^-} \times t_d}, \tag{7}$$





where $J_{NO_3^- \rightarrow HONO}$ is the rate of photolysis of $NO_3^-$ to form HONO, $v_{NO_3^-}$ is the dry deposition rate of $NO_3^-$ during the period $t_d$,
and $f$ is the proportion of the surface exposed to the sun at midday. Here, we suppose that the surfaces involving $NO_3^-$ were
exposed to light by a factor $f = 1/4$, taking mixing height $H = 250\ m$, $v_{NO_3^-} = 5\ cm \cdot s^{-1}$ over $t_d = 24$ h. We use the mean
midday value of $R_{unknown} = 9.77\ \mu g \cdot m^{-3} \cdot h^{-1}$ and $[NO_3^-] = 10.52\ \mu g \cdot m^{-3}$ in spring; and $R_{unknown} = 12.04 \mu g \cdot m^{-3} \cdot h^{-1}$ and
$[NO_3^-] = 3.59\ \mu g \cdot m^{-3}$ in summer. The photolysis rates $J_{NO_3^- \rightarrow HONO}$ derived from Eq. (8) were $5.97 \times 10^{-5}\ s^{-1}$ and
$1.99 \times 10^{-4}\ s^{-1}$ for spring and summer, respectively. These values were in the range $6.2 \times 10^{-6}$ to $5.0 \times 10^{-4}$ obtained in a
previous study (Ye et al., 2017), which indicated that particulate nitrate photolysis could be likely source for the missing
daytime additional HONO formation in spring and summer. The variability of $J_{NO_3^- \rightarrow HONO}$ may be caused by chemical
composition, acidity, light-absorbing constituents, and the optical and other physical properties of aerosols.

### 3.5 Parameterization of HONO

Through an empirical parameterized formula, we can explore an accurate parameterization method for HONO, discuss the
main control factors for the HONO concentration and its chemical behavior, and quantify its main sources and key kinetic
parameters. As mentioned in Sect. 3.1, the HONO/$NO_x$ ratio is better than HONO/$NO_2$ as an indicator of HONO generation.
In another study (Elshorbany et al., 2012), data were collected from 15 field observations all over the world to establish the
correlation between the HONO/$NO_x$ ratio and the HONO concentration in global models. Therefore, we applied this method
in this study to parameterize the HONO concentration. As shown in Fig. 11, the HONO/$NO_x$ ratios in the four seasons were
close to the calculated value (0.02). However, there were seasonal variations in the slope, showing a maximum in summer
($2.60 \times 10^{-2}$), followed by autumn ($2.06 \times 10^{-2}$), and a minimum in winter ($1.59 \times 10^{-2}$). Except for in spring, HONO
showed good correlation with $NO_x$, with $R^2$ values ranging from 0.8972 to 0.9621. Therefore, we used slopes of $2.60 \times 10^{-2}$,
$2.06 \times 10^{-2}$, and $1.59 \times 10^{-2}$ to parameterize the HONO concentrations in summer, autumn, and winter, respectively. As for
spring, though only a weak correlation between HONO and $NO_x$ was found, the majority of the HONO/$NO_x$ ratios fluctuated
round a slope of 0.02 because concentrations of $NO_x$ greater than 60 ppb only accounted for 8.83 % of the data. Therefore, a
slope of 0.02 was applied in spring to parameterize the HONO concentration.
As can be seen from Fig. 12, the estimated values are very close to the observed values in the nighttime in autumn. After
sunrise and before noon, the values observed were higher than the estimated values, and this difference gradually increases.
After noon and before sunset, the values observed were still higher than the values estimated, but the difference gradually
decreases. This phenomenon was also found in the daytime in spring and summer, but not in winter. Compared with the
daytime, the estimated values during the nighttime were closer to the observed values in both trend and value in all four
seasons, which further demonstrates that nighttime HONO is mainly produced from the direct vehicle emissions and
heterogeneous reaction of $NO_2$ on the ground or the surfaces of aerosols. Therefore, we should pay much more attention to
simulation in the daytime. We distinguish two main sectors, nighttime and daytime, to analyze the factors affecting the





HONO diurnal variation (Liu, 2017). Although $J$(HONO)×HONO also correlated well with $J$(NO$_2$)×NO$_2$ in all four seasons
in this study and the linear fitting coefficients fluctuated around 0.01 in all four seasons (Fig. S5), bad simulation results
during the daytime were found (Fig. S6) using
$[HONO] = k \times [NO_2] \times J(NO_2)/J(HONO).$    (8)
Where k was the linear fitting coefficient between $J$(HONO)×HONO and $J$(NO$_2$)×NO$_2$. In contrast, excellent simulation
results were found in a previous study using the same formula (Liu, 2017), which suggests that using the same simulation
formula in different regions may obtain greatly varying results. Eq. (8) can be regarded as a combination of [NO$_2$] with
$J$(NO$_2$)/$J$(HONO). $J$(NO$_2$)/$J$(HONO) kept relatively constant (5.48~5.87) in the daytime in four seasons.  Therefore, diurnal
variation of [HONO] simulated by Eq. (8) depended on [NO$_2$] (Fig. S7). Eq. (8) is only suitable for regions where the diurnal
variation of [NO$_2$] is consistent with that of [HONO].
As discussed in Sect. 3.4.2, nitrate photolysis was perhaps the source of HONO in this study. Besides, the difference
between the observed value and the simulated value kept increasing before noon and the difference began to decrease after
noon, which was consistent with nitrate photolysis. Therefore, we take the photolysis of nitrate into the HONO concentration
simulation. The specific formulas for the simulation of spring, summer, autumn and winter as shown as follow:
$HONOspring = 2.00 \times 10^{-2} \times NOx + [NO_3^-] \times J(NO_3\_R)/4$    (9)
$HONOsummer = 2.60 \times 10^{-2} \times NOx + [NO_3^-] \times J(NO_3\_R)$    (10)
$HONOautumn = 2.06 \times 10^{-2} \times NOx + [NO_3^-] \times J(NO_3\_R)$    (11)
$HONOwinter = 1.59 \times 10^{-2} \times NOx + [NO_3^-] \times J(NO_3\_R)/4$    (12)
In this way, the daytime simulation results are significantly improved (Fig. 12). This further demonstrates that the
apportionment of HONO sources is credible. The parameterization described in this work was more reasonable and can be
better used in the future in such coastal sites.
**3.6 Comparison of contributions of HONO and O$_3$ to OH radicals**
Comparing the OH radical production via photolysis of HONO and O$_3$, the effect of the high HONO concentrations in the
daytime on the tropospheric oxidation capacity was evaluated (Ryan et al., 2018). Nitrous acid is considered to be a crucial
source of OH radicals (Lee et al., 2016). As shown in Eq. (12), OH production rates from O$_3$ photolysis ($P_{OH}$(O$_3$)) were
calculated based on [O$_3$], $J$(O$^1$D), and [H$_2$O] (Liu et al., 2019c). Only O($^1$D) atoms produced by the O$_3$ photolysis at UV
wavelengths less than 320 nm (R6) can combine with water to generate OH radicals (R7) in the atmosphere. The absolute
water concentration was derived from temperature and RH. The reaction (R8) rates for N$_2$ is $3.1 \times 10^{-11}$ cm$^3$ molecules$^{-1}$ s$^{-1}$
and for O$_2$ is $4.0 \times 10^{-11}$ cm$^3$ molecules$^{-1}$ s$^{-1}$(Liu et al., 2019a). The net OH formation from HONO was estimated by Eq. (13)





(Su et al., 2008b; Sörgel et al., 2011a; Li et al., 2018a; Atkinson et al., 2004). In addition to the two primary production of
OH radicals mentioned above, there are the reaction of organic and hydro peroxy radicals ($RO_2$ and $HO_2$) with NO, hydrogen
peroxide photolysis and the ozonolysis of alkenes (Hofzumahaus et al., 2009; Gligorovski et al., 2015; Wang et al., 2018).
$P_{OH}(O_3) = 2J(O^1D)[O_3]\phi OH, \quad \phi OH = k_7[H_2O]/(k_7[H_2O] + k_8[M])$ (12)
$O_3 + hv \rightarrow O(^1D) + O_2$ ($hv < 320$ nm) (R6)
$O(^1D) + H_2O \rightarrow 2OH$ (R7)
$O(^1D) + M \rightarrow O(^3P) + M$ ($M$ is $N_2$ or $O_2$) (R8)
$P_{OH}(HONO) = J_{HONO}[HONO] - k_{OH+NO}[NO][OH] - k_{OH+HONO}[HONO][OH]$ (13)
The diurnal patterns of $P$(OH) are shown in Fig. 13. The formation rates of OH from $O_3$ photolysis peaked in midday at
around 0.71 ppb·h$^{-1}$, 5.80 ppb·h$^{-1}$, 2.21 ppb·h$^{-1}$, and 0.48 ppb·h$^{-1}$ for spring, summer, autumn, and winter, respectively. The
variation of $P_{OH}(O_3)$ is consistent with $J(O^1D)$ (Fig. S8), peaking in midday and in summer on a diurnal and a seasonal
timescale, respectively. For summer and autumn, $P_{OH}(HONO)$ had a similar trend as $P_{OH}(O_3)$, peaking at around noon at the
time of the highest $J$(HONO), but this was negligible at sunrise and sunset (Fig. S9). For spring and winter, however,
$P_{OH}(HONO)$ reached a maximum in the morning rush hour caused by the combined influences of high HONO concentration
and high $J$(HONO). A similar result was also found in southwest Spain from mid-November to mid-December 2008 (Sörgel
et al., 2011a).The HONO photolysis contributed significantly more OH than $O_3$ photolysis during the whole daytime in
spring, autumn, and winter. In summer, the HONO photolysis contributed to more OH in the early morning, and although the
$O_3$ photolysis produced more in the afternoon, HONO photolysis had a considerable effect on OH production. A similar
result was also found in Nanjing of eastern China from November 2017 to November 2018(Liu et al., 2019a). These results
show that HONO contributes considerably to the atmospheric oxidizing capacity of the suburban atmosphere of Xiamen.
Although HONO concentrations (average: 0.66 ppb) are much lower than $O_3$ concentrations (average: 35.88 ppb) during
07:00–16:00 LT, daytime HONO photolysis forms significantly more OH than daytime photolysis of $O_3$ in four seasons
except for summer afternoon. Generally, the mean value of $P_{OH}(HONO)$ from 07:00 to 16:00 LT was 1.89 ppb·h$^{-1}$, and the
average $P_{OH}(O_3)$ was 1.14 ppb·h$^{-1}$. A similar result was found in Melbourne, where the peak OH production rate reached
2 ppb·h$^{-1}$ from 0.4 ppb HONO (Ryan et al., 2018). The important role of HONO in the production of OH promotes
photochemical peroxyacetyl nitrate formation (Hu et al., 2020).
**4. Conclusions**
We conducted measurements of HONO in the atmosphere at an IUE supersite in a coastal city of southeastern China in
August, October, and December 2018 and March 2019, finding an average HONO concentration of 0.54 ± 0.47 ppb across



the whole observation period. Concentrations of HONO in spring and summer were higher than in winter and autumn, which
was consistent with seasonal variations in RH. Both higher HONO concentrations in the daytime and the HONO/NO$_x$ ratio
peaking around noon suggested that additional sources of HONO might be related to light. It was found that the contribution
from vehicle exhaust emissions (1.64 %) was higher than that found in most other studies due to the site being surrounded by
several expressways with a large number of passing diesel vehicles. The average nocturnal conversion rate of NO$_2$ to HONO
was 0.47 % h$^{-1}$, which was within the range 0.29–2.40 % h$^{-1}$ found by other studies. The HONO$_{corr}$/NO$_2$ ratio increased with
RH and the concentration of PM$_{2.5}$ during the nighttime, which indicates that nocturnal heterogeneous reactions on the
surfaces of aerosols are the major source of HONO. However, dark heterogeneous formation ($P_{hete}$) was almost negligible in
the daytime, accounting for approximately 8.07 % of known sources across the whole observation period. $R_{unknown}$ made up
at the largest proportion of all sources in summer (79.55 %), autumn (71.51 %), spring (69.67 %), and winter (55.64 %). It
was found that there was a logarithmic relationship between $R_{unknown}$ and particulate nitrate photolysis in four seasons. The
variation of HONO at night can be accurately simulated based on the HONO/NO$_x$ ratio, while $J(NO_3^-\_R) \times pNO_3^-$ or
$1/4 \times (J(NO_3^-\_R) \times pNO_3^-)$ should be considered for daytime simulation. Local tropospheric oxidation capacity was
significantly increased by HONO during 07:00–16:00, providing an OH radical source 1.89 ppb·h$^{-1}$.
**Data availability**
The observation data at this site are available from the authors upon request.
**Authorship Contribution Statement**
Baoye Hu and Jun Duan contributed equally to this work. Baoye Hu and Jun Duan collected the HONO data and analyzed
the data. Baoye Hu wrote the manuscript. Baoye Hu, Jun Duan performed the experiments. Jun Duan and Fang Wu built
equipment of IBBCEEAS. Youwei Hong, Min Qin and Jinsheng Chen revised manuscript. Min Qin, Pinhua Xie and
Jinsheng Chen designed the manuscript. Jinsheng Chen supported funding of observation and research. Lingling Xu,
Mengren Li, Yahui Bian contributed to discussions of results.
**Competing interests**
The authors declare that they have no conflict of interest.
**Acknowledgments**
This study was funded by the Cultivating Project of Strategic Priority Research Program of Chinese Academy of Sciences
(XDPB1903), the National Key Research and Development Program (2017YFC0209400, 2016YFC02005,





2016YFC0112200), the National Natural Science Foundation of China (41575146, 41875154), the FJIRSM&IUE Joint
Research Fund (RHZX-2019-006), the Center for Excellence in Regional Atmospheric Environment, CAS (E0L1B20201),
State Key Laboratory of Environmental Chemistry and Ecotoxicology, Research Center for Eco-Environmental Sciences,
CAS and Xiamen Atmospheric Environment Observation and Research Station of Fujian Province.
**Supplementary information**
Attached please find supplementary information associated with this article.





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



**Figure Captions**

**Figure 1.** Location of Xiamen in China (left) and surroundings of IUE.

**Figure 2.** Time series of relative humidity (RH), temperature (T), $J$(HONO), UV, HONO, $NO_2$, NO, $NO_3^-$, $PM_{2.5}$, $O_3$, and black carbon (BC) in Xiamen, China in August, October, and December 2018, and March 2019. The missing data is mainly due to instrument maintenance.

**Figure 3.** Diurnal variations in HONO concentration on days with and without SLBs.

**Figure 4.** Diurnal variations in (a) HONO, (b) NO (hollow markers and dashed lines) & $NO_x$ (solid markers/lines), (c) HONO/$NO_x$, and (d) $J$($NO_2$). The gray shading indicates nighttime (18:00–06:00, including 18:00).

**Figure 5.** Scatter plots of $NO_2$ versus HONO color coded by $J$($NO_2$). The three dashed lines represent 10 %, 5 %, and 1 % ratios of HONO/$NO_2$. Daytime was 06:00–18:00 LT, including 06:00.

**Figure 6.** Scatter plots of nighttime $HONO_{corr}$/$NO_2$ ratios versus RH. The average top-five $HONO_{corr}$/$NO_2$ in every 5 % RH interval are shown as orange squares, and the error bars show ±1 SD.

**Figure 7.** The correlation between the $NH_3$ concentration and HONO/$NO_2$ ratio (upper) and the correlation between the $NH_3$ concentration and the $NO_3^-$/$NO_2$ (lower) in four seasons. The scatter points were colored by ambient RH values.

**Figure 8.** The correlation between $PM_{2.5}$ and $HONO_{corr}$ (left) and the correlation between $PM_{2.5}$ and $HONO_{corr}$/$NO_2$ (right). The squares depict $PM_{2.5} \geq 35$ μg·m$^{-3}$; all scattered points are from the time when the ratio of $HONO_{corr}$/$NO_2$ reached a pseudo-steady state each night (03:00–06:00 LT).

**Figure 9.** Average diurnal variations of each source (>0) and sink (<0) of HONO in the four seasons.

**Figure 10.** Relationships between the photolysis of particulate nitrate and $R_{unknown}$, colored by BC in spring, summer, autumn, and winter. Red lines and dashed lines represent logarithmic fitting curve and turning point, respectively.

**Figure 11.** The ratio of HONO/$NO_x$ in the four seasons (correlation between the average of $NO_x$ per 10 ppb interval and the average value of HONO).

**Figure 12.** The diurnal variations in the measured values of HONO (black squares), the estimated values of HONO using the parameterized formula (red circles), and the estimated values of HONO using the parameterized formula combined with the main daytime sources (green triangles).

**Figure 13.** Comparison of OH formation by photolysis of HONO and $O_3$ in the four seasons.

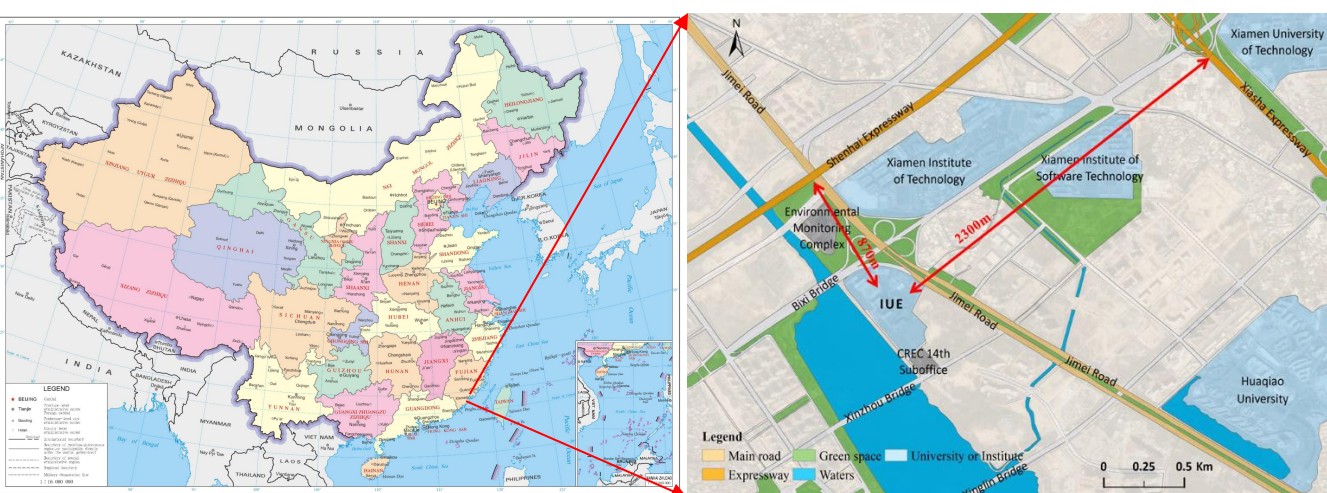

862

**Figure 1.** Location of Xiamen in China (left) and surroundings of IUE (right).

864  Note: The map in the left was directly download from http://bzdt.ch.mnr.gov.cn/, while the map in the right was significantly

865  enriched based on layer download from http://www.rivermap.cn/.





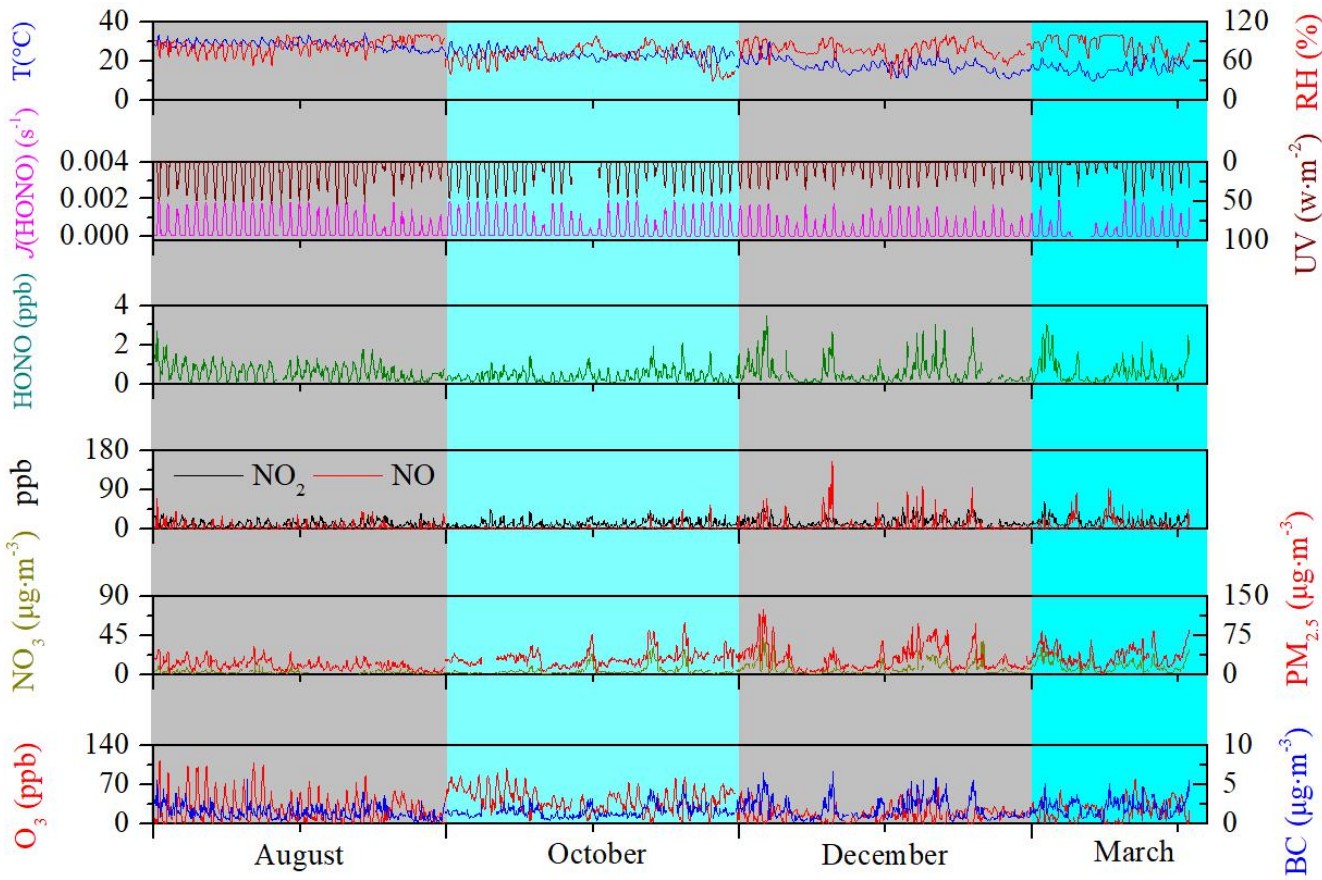

**Figure 2.** Time series of relative humidity (RH), temperature (T), $J$(HONO), UV, HONO, $NO_2$, NO, $NO_3^-$, $PM_{2.5}$, $O_3$, and black carbon (BC) in Xiamen, China in August, October, and December 2018, and March 2019. The missing data is mainly due to instrument maintenance.





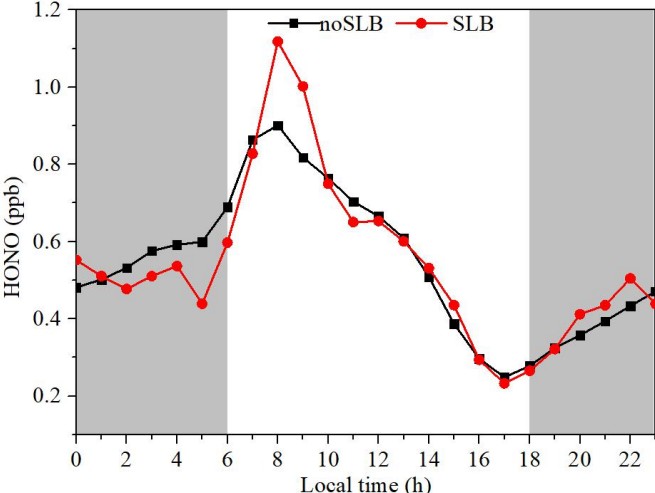


**Figure 3.** Diurnal variations in HONO concentration on days with and without SLBs.




**Figure 4.** Diurnal variations in (a) HONO, (b) NO (hollow markers and dashed lines) & NO$x$ (solid markers/lines), (c) HONO/NOx, and (d) $J(NO_2)$. The gray shading indicates nighttime (18:00–06:00, including 18:00).





**Figure 5.** Scatter plots of NO₂ versus HONO color coded by $J$(NO₂). The three dashed lines represent 10 %, 5 %, and 1 % ratios of HONO/NO₂. Daytime was 06:00–18:00 LT, including 06:00.





**Figure 6.** Scatter plots of nighttime $HONO_{corr}/NO_2$ ratios versus RH. The average top-five $HONO_{corr}/NO_2$ in every 5 % RH interval are shown as orange squares, and the error bars show ±1 SD.





883

884

**Figure 7.** The correlation between the NH$_3$ concentration and HONO/NO$_2$ ratio (upper) and the correlation between the NH$_3$ concentration and the NO$_3^-$/NO$_2$ (lower) in four seasons. The scatter points were colored by ambient RH values.







**Figure 8.** The correlation between $PM_{2.5}$ and $HONO_{corr}$ (left) and the correlation between $PM_{2.5}$ and $HONO_{corr}/NO_2$ (right). The squares depict $PM_{2.5} \geq 35$ µg·m$^{-3}$; all scattered points are from the time when the ratio of $HONO_{corr}/NO_2$ reached a pseudo-steady state each night (03:00–06:00 LT).





**Figure 9.** Average diurnal variations of each source (>0) and sink (<0) of HONO in the four seasons.





894

**Figure 10.** Relationships between the photolysis of particulate nitrate and $R_{unknown}$, colored by $BC$ in spring, summer, autumn, and winter.
Red lines and dashed lines represent logarithmic fitting curve and turning point, respectively.





**Figure 11.** The ratio of HONO/$NO_x$ in the four seasons (correlation between the average of $NO_x$ per 10 ppb interval and the average value of HONO).





**Figure 12.** The diurnal variations in the measured values of HONO (black squares), the estimated values of HONO using the parameterized formula (red circles), and the estimated values of HONO using the parameterized formula combined with the main daytime sources (green triangles).





904

905

**Figure 13.** Comparison of OH formation by photolysis of HONO and $O_3$ in the four seasons.

907





**Tables**

**Table 1.** Overview of the HONO and $NO_x$ average concentrations measured in Xiamen and comparison with other measurements.

**Table 2.** Emission ratios of fresh vehicle plumes $\Delta HONO/\Delta NO_x$.

**Table 3.** Overview of the conversion frequencies from $NO_2$ to HONO in Xiamen and comparisons with other studies.



**Table 1.** Overview of the HONO and NO$_x$ average concentrations measured in Xiamen and comparison with other
measurements.

| Location | Date | HONO (ppb) | | NO$_2$ (ppb) | | NOx (ppb) | | HONO/NO$_2$ | | HONO/NOx | | Reference |
|---|---|---|---|---|---|---|---|---|---|---|---|---|
| | | Day | Night | Day | Night | Day | Night | Day | Night | Day | Night | |
| Xiamen/China (suburban) | Aug.2018-Mar.2019 | 0.63 | 0.46 | 13.6 | 16.3 | 20.9 | 19.9 | 0.061 | 0.028 | 0.046 | 0.024 | This work |
| | Mar.2019(spring) | 0.72 | 0.51 | 18.5 | 17.7 | 28.6 | 24.5 | 0.046 | 0.032 | 0.034 | 0.028 | |
| | Aug.2018(summer) | 0.72 | 0.51 | 11.0 | 15.7 | 16.6 | 18.9 | 0.094 | 0.031 | 0.072 | 0.027 | |
| | Oct.2018(autumn) | 0.50 | 0.33 | 11.4 | 14.3 | 14.1 | 15.1 | 0.060 | 0.023 | 0.048 | 0.022 | |
| | Dec.2018(winter) | 0.61 | 0.52 | 15.8 | 18.3 | 28.0 | 23.1 | 0.036 | 0.026 | 0.023 | 0.022 | |
| Jinan/China (urban) | Sep 2015-Aug 2016 | 0.99 | 1.28 | 25.8 | 31.0 | 40.6 | 46.4 | 0.056 | 0.079 | 0.035 | 0.040 | (Li et al., 2018a) |
| | Sep.-Nov. 2015 (autumn) | 0.66 | 0.87 | 23.2 | 25.4 | 37.5 | 38.0 | 0.034 | 0.049 | 0.022 | 0.034 | |
| | Dec.2015-Feb.2016(winter) | 1.35 | 2.15 | 34.6 | 41.1 | 64.8 | 78.5 | 0.047 | 0.056 | 0.031 | 0.034 | |
| | Mar.-May 2016 (spring) | 1.04 | 1.24 | 25.8 | 35.8 | 36.0 | 47.3 | 0.052 | 0.046 | 0.041 | 0.035 | |
| | Jun.-Aug. 2016 (summer) | 1.01 | 1.20 | 19.0 | 22.5 | 25.8 | 29.1 | 0.079 | 0.106 | 0.049 | 0.060 | |
| Nanjing/China (suburban) | Nov. 2017-Nov. 2018 | 0.57 | 0.80 | 13.9 | 18.9 | 19.3 | 24.9 | 0.044 | 0.045 | 0.036 | 0.041 | (Liu et al., 2019c) |
| | Dec.-Feb. (winter) | 0.92 | 1.15 | 23.1 | 28.4 | 37.7 | 45.5 | 0.038 | 0.040 | 0.025 | 0.029 | |
| | Mar.-May (spring) | 0.59 | 0.76 | 12.9 | 17.4 | 15.9 | 19.1 | 0.049 | 0.048 | 0.042 | 0.046 | |
| | Jun.-Aug. (summer) | 0.34 | 0.56 | 7.7 | 12.5 | 9.1 | 13.5 | 0.051 | 0.048 | 0.045 | 0.046 | |
| | Sep.-Nov. (autumn) | 0.51 | 0.81 | 13.4 | 18.9 | 17.7 | 25.1 | 0.035 | 0.044 | 0.029 | 0.039 | |
| Hongkong/China | Aug.2011(summer) | 0.70 | 0.66 | 18.1 | 21.8 | 29.3 | 29.3 | 0.042 | 0.031 | 0.028 | 0.025 | (Xu et al., 2015) |
| | Nov.2011(autumn) | 0.89 | 0.95 | 29.0 | 27.2 | 40.6 | 37.2 | 0.030 | 0.034 | 0.021 | 0.028 | |
| | Feb.2012(winter) | 0.92 | 0.88 | 25.8 | 22.2 | 48.3 | 37.8 | 0.035 | 0.036 | 0.020 | 0.025 | |
| | May2012(spring) | 0.40 | 0.33 | 15.0 | 14.7 | 21.1 | 19.1 | 0.030 | 0.022 | 0.022 | 0.019 | |
| Guangzhou/China (urban) | Jun.2006 | 2.00 | 3.50 | 30.0 | 20.0 | - | - | 0.067 | 0.175 | - | - | (Qin et al., 2009) |
| Xi'an/China | Jul.-Aug.2015 | 1.57 | 0.51 | 24.7 | 15.4 | - | - | 0.062 | 0.033 | - | - | (Huang et al., 2017) |
| Santiago/Chile (urban) | Mar.-Jun.2005 | 1.50 | 3.00 | 20.0 | 30.0 | 40.0 | 200.0 | 0.075 | 0.100 | 0.038 | 0.015 | (Elshorbany et al., 2009) |
| Rome/Italy (urban) | May-Jun.2001 | 0.15 | 1.00 | 4.0 | 27.2 | 4.2 | 51.2 | 0.038 | 0.037 | 0.024 | 0.020 | (Acker et al., 2006) |
| Kathmandu/Nepal (urban) | Jan.-Feb.2003 | 0.35 | 1.74 | 8.6 | 17.9 | 13.0 | 20.1 | 0.041 | 0.097 | 0.027 | 0.087 | (Yu et al., 2009) |

Note: Night (18:00-6:00, including 18:00, local time); Day (6:00-18:00, including 6:00, local time)
NOx=NO$_2$ (IBBCEAS)+NO (Thermal 42i). IBBCEAS measure both HONO and NO$_2$. The NO$_2$ concentration is always overestimated by
the Thermo Fisher 42i.





**Table 2.** Emission ratios of fresh vehicle plumes ΔHONO/ΔNO$x$.

| Date | Time | ΔNO/ΔNOx | $R^2$ | ΔHONO/ΔNOx (%) |
|---|---|---|---|---|
| 2018/8/1 | 7:00-8:55 | 1.1621 | 0.6897 | 2.17 |
| 2018/8/8 | 5:40-5:55 | 0.8727 | 0.8023 | 2.69 |
| 2018/8/21 | 5:00-5:55 | 0.8571 | 0.7553 | 1.14 |
| 2018/8/22 | 7:20-7:45 | 0.4998 | 0.6151 | 4.76 |
| 2018/8/23 | 5:20-5:55 | 0.7321 | 0.8089 | 2.12 |
| 2018/8/23 | 6:00-6:55 | 0.8321 | 0.6687 | 2.19 |
| 2018/8/31 | 23:35-23:55 | 1.1861 | 0.8130 | 1.18 |
| 2018/10/23 | 1:05-1:25 | 0.9893 | 0.6566 | 1.27 |
| 2018/12/4 | 7:20-7:40 | 0.9594 | 0.8502 | 1.11 |
| 2018/12/10 | 11:00-11:15 | 0.8778 | 0.6735 | 1.79 |
| 2018/12/11 | 0:00-0:50 | 0.9424 | 0.6972 | 0.58 |
| 2018/12/11 | 1:25-1:55 | 0.8492 | 0.8237 | 1.26 |
| 2018/12/11 | 2:50-3:55 | 0.7405 | 0.7520 | 2.87 |
| 2018/12/11 | 4:00-4:55 | 0.9652 | 0.7686 | 2.12 |
| 2018/12/11 | 5:45-6:35 | 1.0243 | 0.6566 | 0.84 |
| 2018/12/11 | 6:40-7:40 | 0.9992 | 0.7067 | 1.59 |
| 2018/12/11 | 8:15-8:55 | 0.8333 | 0.6820 | 1.89 |
| 2018/12/13 | 7:00-8:50 | 0.8263 | 0.8127 | 1.02 |
| 2018/12/13 | 9:10-9:45 | 0.7235 | 0.7776 | 1.01 |
| 2018/12/16 | 7:00-7:55 | 0.7523 | 0.8939 | 0.98 |
| 2018/12/18 | 7:35-8:10 | 0.7046 | 0.7110 | 1.15 |
| 2018/12/20 | 22:50-23:10 | 0.9811 | 0.7736 | 0.97 |
| 2018/12/21 | 0:45-1:15 | 1.0029 | 0.8914 | 1.54 |
| 2018/12/22 | 6:40-7:35 | 1.0194 | 0.7010 | 2.36 |
| 2018/12/22 | 7:40-8:05 | 0.9932 | 0.7831 | 2.94 |
| 2018/12/25 | 21:00-22:10 | 0.9573 | 0.8857 | 1.64 |
| 2018/12/26 | 3:50-4:15 | 1.167 | 0.6540 | 1.39 |
| 2018/12/26 | 6:45-7:45 | 0.9971 | 0.8463 | 0.92 |
| 2018/12/26 | 7:55-8:25 | 0.9714 | 0.6919 | 2.95 |
| 2018/12/27 | 4:50-5:30 | 0.9365 | 0.7265 | 0.76 |
| 2019/3/6 | 7:30-8:05 | 1.0309 | 0.8283 | 0.74 |
| 2019/3/9 | 7:50-8:05 | 0.9933 | 0.9203 | 0.24 |
| 2019/3/9 | 12:00-12:55 | 0.9627 | 0.6444 | 0.51 |
| 2019/3/18 | 6:35-8:35 | 1.0382 | 0.6967 | 3.14 |




**Table 3.** Overview of the conversion frequencies from NO$_2$ to HONO in Xiamen and comparisons with other studies.

| Location | Date | Conversion rate (% h-1) | Reference |
|---|---|---|---|
| Xiamen/China | Aug.2018-Mar.2019 | 0.47 | This study |
| | Mar.2019(spring) | 0.47 | |
| | Aug.2018(summer) | 0.55 | |
| | Oct.2018(autumn) | 0.48 | |
| | Dec.2018(winter) | 0.37 | |
| Xinken/China | Oct.-Nov.,2004 | 1.60 | (Su et al., 2008c) |
| Jinan/China | Sep.,2015-Aug.,2016 | 0.68 | (Li et al., 2018a) |
| | Mar.-May 2016(spring) | 0.43 | |
| | Jun.-Aug. 2016(summer) | 0.69 | |
| | Sep.-Nov. 2015(autumn) | 0.75 | |
| | Dec.2015-Feb. 2016(winter) | 0.83 | |
| Guangzhou/China | Jun.,2006 | 2.40 | (Li et al., 2012b) |
| Spain | Nov.-Dec.,2008 | 1.50 | (Sörgel et al., 2011a) |
| Beijing/China | Sep.2015-July 2016 | 0.80 | (Wang et al., 2017) |
| | Apr.-May, 2016 (spring) | 0.50 | |
| | Jun.-Jul., 2016 (summer) | 1.00 | |
| | Sep.-Oct. 2015 (autumn) | 0.90 | |
| | Jan.2016 (winter) | 0.60 | |
| Shandong/China | Nov.2013-Jan.2014 | 0.29 | (Wang et al., 2015) |
| Shanghai/China | Aug.2010-Jun.2012 | 0.70 | (Wang et al., 2013) |
| Eastern Bohai Sea/China | Oct.-Nov., 2016 | 1.80 | (Wen et al., 2019a) |
| Hongkong/China | Aug.2011-May, 2012 | 0.52 | (Xu et al., 2015) |
| Kathmandu/South Asia | Jan.-Feb.,2003 | 1.4 | (Yu et al., 2009) |
