# Peer review of "Exploration of the atmospheric chemistry of nitrous acid in a coastal"

_Atmospheric Chemistry and Physics, 2021_

## Author Response (AR1)

**RC1 Anonymous Referee#2**

Hu et al. performed seasonal field observations of HONO in Xiamen, China in August, October, and December 2018, and March 2019, along with measurements of trace gases, aerosol compositions, photolysis rate constants (J), and meteorological parameters. The result shows that the average observed concentration of HONO is $0.54 \pm 0.47$ ppb. Vehicle exhaust emissions is an important source of HONO. By considering the influence factors on HONO formation, further explains nighttime heterogeneous conversion of $NO_2$ to HONO. The daytime unknown sources are likely to be related to light, it is find that there is a logarithmic relationship between $P_{unknown}$ and particulate nitrate photolysis in four seasons. Then, the different parameters of nitric acid during photolysis are discussed, and using the resulting parameters for simulation. The simulated results are compared with the observed values. Finally, daytime HONO photolysis forms significantly more OH than daytime photolysis of $O_3$ in four seasons except for summer afternoon, further explains the importance of HONO in atmospheric chemistry. The manuscript can be considered to be accepted after addressing the following comments. The language can be further improved to make it easier for readers to follow up.

*Response: Thanks for your valuable comments and positive feedback. We have corrected this manuscript according to your suggestion. Below are the point-to-point responses to general and specific comments.*

Manuscript should be proofread before resubmission to avoid minor errors, such as Line 32, add "ppbh$^{-1}$" after 2.05; Line 100, "gas" should be "gases"; Line 173, "Figure 2" should be "Fig. 2", I think it should be uniform throughout the text.

*Response: Thanks for your responsible review. Manuscript has been proofread to avoid minor errors. "ppb·h$^{-1}$" has been added after 2.05 in Line 32, "gas" has been changed into "gases" in Line 100, and "Figure 2" has been changed into "Fig. 2 in Line 173".*

Lines 107-108. the English usage in the statement of " The surrounding soil is used for green not for agriculture " is not understandable and the sentence should be rephrased.

*Response: Thanks for your careful working. This sentence has been changed into "The surrounding soil is used for landscape greening not for agricultural production".*

Lines 118, There should be more detailed information in the parentheses, such as model number, manufacturer, and region.

*Response: Thanks for your careful working. Model number, manufacturer, and region have been added in the parentheses like this: (QE65000, Ocean Optics Inc., USA).*

Lines 158-159, the sentence is grammatically incorrect. And correct other similar mistakes in this manuscript.

*Response: Thanks for your careful working. This sentence and similar mistakes have been corrected. This sentence has been changed into "The $O_3$ concentration was determined by*

*ultraviolet photometric analyzer [Model 49i, Thermo Environmental Instrument (TEI) Inc.], and the limit of the instrument is 1.0 ppb".*

Lines 254, delete an "in".

*Response: Thanks for your careful working. An "in" has been deleted in Line 254.*

Lines 268-272, The concurrence of both "(2) short duration air masses (<2 h)" and "(5) NO/NOx > 0.50" may result in inaccuracies, the emission factors need to be calculated in fresh plumes, wider constraints do not guarantee that most of the calculated NO comes from vehicle emissions.

*Response: Thanks for your careful working. $\Delta NO/\Delta NOx > 0.85$ was adopted to indicate fresh plumes(Liu et al., 2019a). $\Delta NO/\Delta NOx > 0.80$ was adopted to indicate fresh plumes(Xu et al., 2015a). $\Delta NO/\Delta NOx > 0.7$ was adopted to indicate fresh plumes(Li et al., 2018). Therefore, $\Delta NO/\Delta NOx > 0.85$ has been adopted to characterize fresh plumes in this study to replace $\Delta NO/\Delta NOx > 0.50$ to guarantee that most of the calculated NO comes from vehicle emissions.*

Lines 382-384, dose "H" is the mixing layer height? If we use the mixing layer heights (1074.4 m) in spring and =0.2 cms⁻¹ to calculate the dry deposition time, the dry deposition time is 14.9h, that's longer than HONO's life span. So I think the author should reconsider the meaning of "H".

*Response: Thanks for your valuable suggestions. "H" is the mixing layer height. A mixing height of 1000 m was used to parameterize $L_{dep}$ (Sörgel et al., 2011; Yu et al., 2021; Su et al., 2008). 500 m were used to parameterize $L_{dep}$ (Xue et al., 2020; Zhang et al., 2019) because the solar radiation reduces significantly. Due to the rapid photolysis of HONO at daytime during its vertical transport, most of HONO can not reach the height above 200 m(Li et al., 2018; Alicke et al., 2002; Liu et al., 2019a). Therefore, the mixing layer height 200 m was used to parameterize $L_{dep}$ in four seasons.*

**RC2 Anonymous Referee#1**

The manuscript "Exploration of the atmospheric chemistry of nitrous acid in a coastal city of southeastern China: Results from measurements across four seasons" by Baoye Hu et al. reports year-long observations of HONO together with gaseous, particulate, and meteorological parameters which are relevant for investigating HONO sources. The manuscript adds valuable information on HONO concentration level and its temporal variation under costal condition. I have reviewed the old version which submitted to ACP in 2020. The quality of the new manuscript is significantly improved. I only have the following minor suggestions before it is accepted for publication.

*Response: Thanks for your valuable comments and positive feedback. We have corrected this manuscript according to your suggestion. Below are the point-to-point responses to general and specific comments.*

Line 31-32: an unit is missing here, should be "2.05 ppb h$^{-1}$ in winter".

*Response: Thanks for your constructive comments. "ppb·h$^{-1}$" has been added after 2.05.*

Line 209-212: It's far fetched to say the concentration of sea salt during the daytime (2.91 µg·m$^{-3}$) is higher than that during the night (2.73 µg·m$^{-3}$). From my side, these levels are similar. I doubt if this small difference in sea salt could have large effect on the contrasting HONO levels between daytime and nighttime. Figure 4 shows that NO$_X$ concentration in the daytime is higher than in the nighttime. The higher HONO in the daytime is more likely due to the higher NO$_X$ or nitrate photolysis as you discussed in following section.

*Response: Thanks for your valuable suggestions. The effect of sea salt and SLBs on the contrasting HONO levels between daytime and nighttime has been deleted from the manuscript. This sentence "The higher HONO in the daytime is likely due to the higher NOx or nitrate photolysis as discussed in following section." has been added in the manuscript.*

Line 214: SLB should be defined here.

*Response: Thanks for your careful working. The effect of sea salt and SLBs on the contrasting HONO levels between daytime and nighttime has been deleted from the manuscript.*

Line 216-224 and line 247-256: The two paragraphs are overlapping. I suggest to integrate the two paragraphs. Figure 5 shows the correlation between NO$_2$ and HONO, it is better to also display the correlation between NO$_X$ and HONO here.

*Response: Thanks for your suggestions. The paragraph from Line 216-224 mainly mentioned why the ratio of HONO/NOx was used and compared with other studies, while the paragraph from Line 247-256 mainly mentioned the diurnal variations of HONO/NOx and the effect of light on HONO formation. Therefore, these two paragraphs have different focus. It is more logical to separately write.*
*Figure 5 showed the correlation between NO$_2$ and HONO color coded by J(NO$_2$), which was used to investigate HONO formation during the daytime is more possibly related to light or Reaction (R5). Therefore, the correlation between NOx and HONO did not display here.*

Line 251: "the photolysis of NO$_2$" should be changed into "the photolysis rate constant of NO2".

*Response: Thanks for your careful working. "The photolysis of NO$_2$" has been changed into "the photolysis rate constant of NO$_2$".*

Line 255-256: "which indicates that HONO formation during the daytime is controlled by light rather than Reaction (R5)." It is hasty to draw this conclusion here. It's better to say "HONO formation during the daytime is more possible to relate to light than Reaction (R5)."

*Response: Thanks for your careful working. This sentence has been changed into "HONO formation during the daytime is more possible to relate to light than Reaction (R5)".*

Line 254: "correspondence" should be changed into "correlation".

*Response: Thanks for your careful working. "Correspondence" has been changed into "correlation".*

Line 269-270: How is the duration of air masses been determined? How do you acquire it?

*Response: Thanks for your careful working. The meaning of the text is that the air mass meets the conditions UV < 10 W·m$^{-2}$, HONO correlating well with NOx (R$^2$ > 0.60, P < 0.05), NOx > 20 ppb (highest 25 % of NOx value), ΔNO/ΔNOx > 0.85, and the duration of the air mass cannot exceed 2 h (Liu et al., 2019b; Xu et al., 2015b), which was based on the following two reasons. Firstly, fresh air mass should be short time. An Air mass with high NOx lasting long could not be a local fresh air mass, but an aged air mass transporting from high NOx region, such as city region. Secondly, if the duration of air mass was too long, the HONO observed was easily affected by secondary production, which would overestimate vehicle emission.*

Line 273-277: ΔNO/ΔNOx and ΔHONO/ΔNOx should be clearly defined.

*Response: Thanks for your careful working. ΔNO/ΔNOx and ΔHONO/ΔNOx have been clearly defined in the manuscript. ΔNO/ΔNOx and ΔHONO/ΔNOx represent the linear slope of NO with NOx, and HONO with NOx, respectively.*

Section 3.5: first of all, there are various assumptions on HONO production pathways been made in the previous sections. It would be better to provide a full picture on how large of each contribution to the HONO formation.

*Response: Thanks for your valuable suggestions. Budget analysis of HONO provide a semi-quantitative understanding of source contribution to the HONO formation. However, it is difficult to provide a full picture on how large of each contributor to the HONO formation due to a lack of atmospheric simulation smog chamber facility. Therefore, parameterization of HONO was used to make up for the shortcomings.*

Line 412-413: "We will discuss… in the next section". Do you mean "in this section"?

*Response: Thanks for your careful working. "in the next section" has been changed into "in this section".*

Line 448-449: "As shown in Fig. 11, the HONO/NOx ratios in the four seasons were close to the calculated value (0.02)". What is the calculated value? Do you mean "the calculated value by Elshorbany et al., 2012?

*Response: Thanks for your careful working. The HONO/NOx ratios should be changed into ΔHONO/ΔNOx ratios, which represents the linear slope of HONO with NOx. The linear slope of HONO with NOx can directly obtained from Fig.11 rather than the calculated value by Elshorbany et al., 2012.*

Fig. 5 and Fig. 11: The correlation between HONO and NO₂ in spring is better than that between HONO and NOx. Why?

*Response: The correlation between HONO and NO$_2$ (0.412, P<0.01) in spring is better than that between HONO and NOx (0.257, P<0.01). The correlation coefficients between HONO and NO$_2$ were 0.376 (P<0.01), 0.487 (P<0.01), and 0.665 (P<0.01) for summer, autumn, and winter, respectively. The correlation coefficients between HONO and NOx were 0.588 (P<0.01),0.597 (P<0.01), and 0.853 (P<0.01) for summer, autumn, and winter, respectively. Therefore, the correlation between HONO and NO$_2$ was only better than that between HONO and NOx for spring. Spring has frequent rains with highest monthly average rainfall (6.99 mm), followed by summer (4.70 mm), winter (0.52 mm), and autumn (0.25 mm). HONO correlates better with NOx (0.642, P<0.01) than with NO$_2$ (0.494, P<0.01) when we only choose the days without rains due to significant increase correlation between HONO and NO from 0.098\* to 0.630\*\*.*

**References**

Alicke, B., Platt, U., and Stutz, J.: Impact of nitrous acid photolysis on the total hydroxyl radicalbudget during the Limitation of Oxidant Production/Pianura PadanaProduzione di Ozono study in Milan, J. Geophys. Res., 107, 10.1029/2000JD000075, 2002.

Li, D., Xue, L., Wen, L., Wang, X., Chen, T., Mellouki, A., Chen, J., and Wang, W.: Characteristics and sources of nitrous acid in an urban atmosphere of northern China: Results from 1-yr continuous observations, Atmos. Environ., 182, 296-306, 10.1016/j.atmosenv.2018.03.033, 2018.

Liu, Y., Nie, W., Xu, Z., Wang, T., Wang, R., Li, Y., Wang, L., Chi, X., and Ding, A.: Semi-quantitative understanding of source contribution to nitrous acid (HONO) based on 1 year of continuous observation at the SORPES station in eastern China, Atmospheric Chemistry and Physics, 19, 13289-13308, 10.5194/acp-19-13289-2019, 2019a.

Liu, Y., Nie, W., Xu, Z., Wang, T., Wang, R., Li, Y., Wang, L., Chi, X., and Ding, A.: Semi-quantitative understanding of source contribution to nitrous acid (HONO) based on 1 year of continuous observation at the SORPES station in eastern China, Atmos. Chem. Phys., 19, 13289-13308, 10.5194/acp-19-13289-2019, 2019b.

Sörgel, M., Regelin, E., Bozem, H., Diesch, J. M., Drewnick, F., Fischer, H., Harder, H., Held, A., Hosaynali-Beygi, Z., Martinez, M., and Zetzsch, C.: Quantification of the unknown HONO daytime source and its relation to NO2, Atmospheric Chemistry and Physics, 11, 10433-10447, 10.5194/acp-11-10433-2011, 2011.

Su, H., Cheng, Y. F., Shao, M., Gao, D. F., Yu, Z. Y., Zeng, L. M., Slanina, J., Zhang, Y. H., and Wiedensohler, A.: Nitrous acid (HONO) and its daytime sources at a rural site during the 2004 PRIDE-PRD experiment in China, Journal of Geophysical Research, 113, 10.1029/2007jd009060, 2008.

Xu, Z., Wang, T., Wu, J., Xue, L., Chan, J., Zha, Q., Zhou, S., Louie, P. K. K., and Luk, C. W. Y.: Nitrous acid (HONO) in a polluted subtropical atmosphere: Seasonal variability, direct vehicle emissions and heterogeneous production at ground surface, Atmospheric Environment, 106, 100-109, 10.1016/j.atmosenv.2015.01.061, 2015a.

Xu, Z., Wang, T., Wu, J., Xue, L., Chan, J., Zha, Q., Zhou, S., Louie, P. K. K., and Luk, C. W. Y.: Nitrous acid (HONO) in a polluted subtropical atmosphere: Seasonal variability, direct vehicle emissions and heterogeneous production at ground surface, Atmos. Environ.,

10.1016/j.atmosenv.2015.01.061, 2015b.

Xue, C., Zhang, C., Ye, C., Liu, P., Catoire, V., Krysztofiak, G., Chen, H., Ren, Y., Zhao, X., Wang, J., Zhang, F., Zhang, C., Zhang, J., An, J., Wang, T., Chen, J., Kleffmann, J., Mellouki, A., and Mu, Y.: HONO Budget and Its Role in Nitrate Formation in the Rural North China Plain, Environ Sci Technol, 54, 11048-11057, 10.1021/acs.est.0c01832, 2020.

Yu, Y., Cheng, P., Li, H., Yang, W., Han, B., Song, W., Hu, W., Wang, X., Yuan, B., Shao, M., Huang, Z., Li, Z., Zheng, J., Wang, H., and Yu, X.: Budget of nitrous acid (HONO) and its impacts on atmospheric oxidation capacity at an urban site in the fall season of Guangzhou, China, Atmos. Chem. Phys. Discussion, 10.5194/acp-2021-178, 2021.

Zhang, W., Tong, S., Ge, M., An, J., Shi, Z., Hou, S., Xia, K., Qu, Y., Zhang, H., Chu, B., Sun, Y., and He, H.: Variations and sources of nitrous acid (HONO) during a severe pollution episode in Beijing in winter 2016, Sci Total Environ, 648, 253-262, 10.1016/j.scitotenv.2018.08.133, 2019.